# nD-RoPE: A Generalized RoPE for $n$-Dimensional Position Embedding

**Boyang Li** [1]  **Yulin Wu** [2]  **Sizhe Xu** [1]  **Nuoxian Huang** [3]  **Zhonghang Yuan** [4]  **Shangyi Guo** [1]  **Shu Yang** [1]
**Takahiro Yabe** [1]

## Abstract

Rotary Position Embedding (RoPE) is widely adopted in Transformer models, yet its extension to high-dimensional domains lacks a unified theoretical formulation. Most existing approaches either apply rotations independently along each axis or empirically mix frequencies, which limits cross-dimensional interactions and yields direction-dependent representations. To address these limitations, we propose *nD-RoPE*, a decomposition-free generalization of RoPE to arbitrary dimensions. From a translation-invariant formulation in continuous Hilbert space, we derive a spectral condition for isotropy that requires treating positions and frequencies as coupled $n$-dimensional vectors. We instantiate this formulation with a multi-scale regular-simplex wavevector design, which provides non-degenerate spatial coverage and a symmetric, directionally balanced second-order response. Experiments across images, videos, and point clouds demonstrate consistent performance gains and improved generalization in high-dimensional settings.

## 1. Introduction

Understanding spatial relations plays a crucial role in constructing an internal model of the world, enabling intelligent systems to organize knowledge within complex, high-dimensional cognitive maps (Tolman, 1948; Gornet & Thomson, 2024). From a modeling perspective, spatial relations can be naturally formulated as learning pairwise interactions between positions (Gao et al., 2019), which aligns well with the self-attention mechanism. Accordingly, the Transformer architecture, built upon self-attention, has demonstrated a strong ability to capture complex relational

structures in high-dimensional spaces, a property believed to underlie its success across a wide range of tasks (Reinauer et al., 2021).

However, this relational modeling ability does not by itself encode position, since self-attention is permutation-invariant and cannot distinguish between different spatial arrangements unless positional information is explicitly provided. As a result, substantial research has focused on enhancing position embedding to enable Transformers to better capture this information (Li et al., 2021; Raffel et al., 2020; Press et al., 2021). Among these advancements, RoPE (Su et al., 2024) has gained significant attention. Owing to its strong performance in long-context modeling, RoPE has been adopted in several large language models, including LLaMA series (Touvron et al., 2023; Grattafiori et al., 2024) and Qwen series (Wang et al., 2024; Yang et al., 2025). Nevertheless, RoPE was originally designed for language tasks, and its extension to higher-dimensional spaces faces inherent limitations. In particular, relative positions become increasingly complex in higher-dimensional domains: for example, in 2D images, a positional encoding must jointly represent horizontal and vertical offsets rather than a single sequential distance (Shaw et al., 2018). While several efforts have adapted RoPE to specific high-dimensional settings—spanning images, videos, spatial representations, 3D geometry, and multi-modal inputs (Heo et al., 2024; Wei et al., 2025; Mai et al., 2020; Ji et al., 2025; Lu et al., 2024)—a unified framework for extending RoPE beyond axis-wise constructions is still lacking.

When extending positional encoding beyond 1D, a common approach is to decompose positions into independent axes and encode each dimension separately. However, this axis-wise formulation implicitly assumes that multidimensional displacements can be decomposed without accounting for their geometric coupling. Consider a diagonal shift: modeling it as independent horizontal and vertical rotations artificially fragments a single coherent displacement, disrupting cross-dimensional interactions and leading to inconsistent relative phases in attention computation. This reveals a deeper principle: position should not be disassembled, but instead be encoded as a unified vector (see Fig. 1(a)), regardless of dimensionality. Guided by this principle, we propose an n-dimensional formulation that preserves the intrinsic

[1]New York University, New York, NY, USA [2]Peking University, Beijing, China [3]Imperial College London, London, UK [4]University of Science and Technology of China, Hefei, China. Correspondence to: Takahiro Yabe <takahiroyabe@nyu.edu>.

*Proceedings of the 43rd International Conference on Machine Learning*, Seoul, South Korea. PMLR 306, 2026. Copyright 2026 by the author(s).

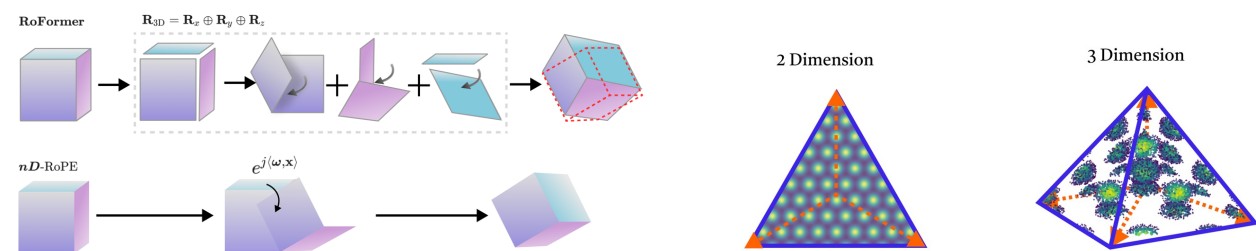

*(a)* Unified $n$-dimensional position–frequency interaction.      *(b)* Regular simplex wave-vector construction.

*Figure 1.* (**a**) *Axis-wise vs. unified position embedding.* Top: conventional axis-wise constructions decompose a displacement into independent 1D components, fragmenting a coherent spatial transformation and introducing directional bias. Bottom: nD-RoPE treats positions as unified $n$-dimensional vectors, preserving cross-dimensional geometric consistency. Positions $\mathbf{x}$ and wave vectors $\boldsymbol{\omega}$ interact through a single rotation $e^{j\boldsymbol{\omega}^\top\mathbf{x}}$. (**b**) *Isotropic wave-vector construction.* Wave vectors are defined by the centroid-to-vertex directions of a regular simplex, whose combined responses yield directionally balanced positional encoding.

geometry of relative positions.

Building on a Fourier-based formulation of translation-invariant positional interactions, we introduce nD-RoPE. It defines rotations directly in $n$-dimensional space through the inner product between position and wave vectors, both treated as $n$-dimensional entities (e.g., $\mathbf{x} = (x_1, x_2, x_3)$ and $\boldsymbol{\omega} = (\omega_1, \omega_2, \omega_3)$, with rotation given by $e^{j\boldsymbol{\omega}^\top\mathbf{x}}$). This yields a single formulation that applies uniformly across dimensionalities. For wave-vector selection, we adopt a regular simplex configuration guided by non-degenerate coverage and geometric symmetry (Conway & Sloane, 2013) (see Fig. 1(b)). As a result, our design avoids the directional bias inherent in axis-wise schemes and naturally captures relative geometry in high-dimensional data.

To evaluate the proposed approach, we conduct experiments across a diverse set of benchmarks spanning multiple dimensions. Across all settings, we compare our method with existing positional embedding techniques. The experimental results demonstrate that the proposed nD-RoPE consistently improves state-of-the-art self-attention models while preserving a strong extrapolation capability across different dimensionalities and data modalities. The contributions of this paper are summarized as follows:

- **Unified n-dimensional RoPE formulation.** We propose *nD-RoPE*, a principled generalization of Rotary Position Embedding to arbitrary dimensions. By deriving rotary modulation from translation-invariant positional interactions, nD-RoPE directly encodes relative positions in the full $n$-dimensional space rather than decomposing them along coordinate axes.

- **Simplex-based isotropic wave-vector design.** We introduce a regular-simplex wave-vector construction guided by non-degenerate coverage and geometric symmetry. This deterministic design provides balanced non-axis-aligned frequency directions and avoids the

directional bias of axis-wise RoPE variants.

- **Comprehensive empirical validation.** We evaluate nD-RoPE across several high-dimensional settings, including images, videos, and point clouds, showing consistent improvements in in-domain performance, rotational robustness, and resolution or density extrapolation over existing positional embedding methods.

## 2. Related Work

Extending RoPE to high-dimensional domains has received increasing attention in recent research. A straightforward approach is to flatten high-dimensional inputs into 1D sequences to apply standard RoPE. However, this naive serialization disrupts the intrinsic spatiotemporal structure, limiting performance even with indexing refinements (Wang et al., 2024). To preserve spatial structure, Axial RoPE (Su et al., 2024) and its variants (Ma et al., 2025; Wei et al., 2025) decompose the position vector into independent components, applying rotary embeddings separately along spatial or temporal axes. While effective for axis-aligned dependencies, this decomposition inherently limits cross-dimensional interactions and introduces directional bias.

Beyond empirical extensions, recent work has explored more principled generalizations of RoPE. Lie group–based formulations introduce learnable generators to enrich positional interactions (Schenck et al., 2025; Ostmeier et al., 2025), but typically rely on commuting generators, limiting holistic geometric coupling. Rethinking RoPE introduces cross-dimensional interactions through learnable transformations (Liu & Zhou, 2025), at the cost of increased complexity and reduced interpretability. A more unified perspective, exemplified by RoPE-Mixed (Heo et al., 2024), treats the coordinate vector as a whole and enables isotropic modeling without privileging individual axes, but lacks a rigorous theoretical foundation for frequency selection in

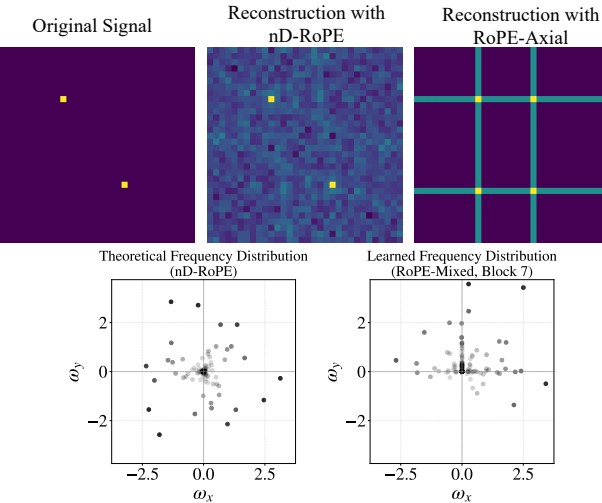

*Figure 2.* **Top:** NUFT reconstruction of impulse signals using nD-RoPE and RoPE-Axial. Axis-aligned artifacts are clearly visible for axis-wise embedding, while nD-RoPE yields sharp and isotropic reconstructions. **Bottom:** Frequency distributions of learned RoPE-Mixed and theoretical nD-RoPE, illustrating potential frequency collapse and anisotropy in learnable variants versus structured multi-scale coverage in nD-RoPE.

higher dimensions.

Theoretically, RoPE can be viewed as a specific instantiation of Fourier feature mappings, which enable coordinate-based networks to represent high-frequency functions (Tancik et al., 2020). From this perspective, FoPE (Hua et al., 2025) analyzes spectral distortion in RoPE and improves 1D length extrapolation via Fourier-based correction, but it does not address the construction of isotropic positional encodings in arbitrary-dimensional spaces. Standard Fourier feature methods typically rely on random Gaussian sampling (Rahimi & Recht, 2007), which provides no guarantee of uniform frequency coverage and can lead to unstable representations in high-dimensional spaces. Conversely, conditional positional encodings (Chu et al., 2021) and NeRF-based methods (Mildenhall et al., 2021) adopt axis-aligned frequency grids, imposing direction-dependent structures. As a result, existing constructions span random sampling, 1D spectral correction, and anisotropic grids, but lack a deterministic and geometrically symmetric solution that scales to arbitrary dimensions.

## 3. Motivational Analysis

The above observations suggest that an effective positional embedding for multi-dimensional spaces should encode relative positions along non-axis-aligned directions and provide uniform coverage across all directions to achieve isotropy in Euclidean space. We analyze how existing RoPE variants violate these criteria and show how nD-RoPE addresses both limitations. For clarity, we focus on the 2D setting; corresponding 3D results are provided in Appendix D.1.

### 3.1. Directional Bias in Axis-Wise Positional Embeddings

Axis-wise positional embeddings model each coordinate independently, e.g., by factorizing the basis into $\exp(i\omega_x x)$, $\exp(i\omega_y y)$, and $\exp(i\omega_z z)$. This restricts the frequency basis to axis-aligned components and biases the positional representation toward coordinate directions. The limitation becomes evident in the non-uniform Fourier transform (NUFT) reconstruction task: reconstructed impulses exhibit pronounced grid-like artifacts along the coordinate axes, consistent with observations in (Heo et al., 2024). These artifacts indicate that oblique frequencies, such as diagonal directions, are poorly represented, leaving large portions of the spectrum underutilized.

nD-RoPE instead treats the frequency vector $\boldsymbol{\omega}$ and position vector $\boldsymbol{x}$ jointly, forming bases $\exp(i\boldsymbol{\omega}^\top \boldsymbol{x})$, where $\boldsymbol{\omega} \in \mathbb{R}^n$ is sampled from a multi-scale simplex. By sampling frequency vectors across directions rather than only along coordinate axes, this construction provides more uniform directional coverage. As shown in the top panel of Fig. 2, NUFT reconstructions with nD-RoPE recover impulse signals more isotropically, without the axis-aligned artifacts observed in axis-wise positional embeddings.

### 3.2. Frequency Collapse and Anisotropy in Learnable RoPE Variants

The generalization ability of RoPE-Mixed suffers from its uneven directional distribution of learned frequencies. We visualize the frequency spectra of RoPE-Mixed (Heo et al., 2024) at a representative transformer block, where positional embedding takes the form $e^{i(\omega_x x + \omega_y y)}$ with trained frequency parameters $\boldsymbol{\omega}$. As shown in the bottom-right panel of Fig. 2, although the frequencies are initialized to span multiple scales, optimization may drive the learned $\boldsymbol{\omega}$ vectors to collapse into irregular, low-frequency clusters with a highly anisotropic distribution. Consequently, large regions of the frequency spectrum remain underutilized, limiting generalization and making the representation unstable.

In contrast, nD-RoPE constructs positional embeddings by sampling frequency vectors from a regular simplex at each scale, with an additional random rotation, thereby guaranteeing multi-scale coverage and isotropic geometry. As illustrated in the bottom-left panel of Fig. 2, nD-RoPE produces structured concentric shells in the frequency domain, where the $\boldsymbol{\omega}$ vectors form multi-scale spherical shells that preserve geometric regularity. Below, we explain how nD-RoPE achieves these observed effects through a theoretical and formalized lens.

# 4. Method

In this section, we present nD-RoPE, a positional embedding framework for arbitrary-dimensional inputs. Our formulation treats the input position $\mathbf{x} \in \mathbb{R}^n$ and the wave vector $\boldsymbol{\omega} \in \mathbb{R}^n$ jointly as $n$-dimensional vectors, without axis-wise decomposition. Positional information is encoded through complex exponentials of the form $e^{j\boldsymbol{\omega}^\top \mathbf{x}}$, enabling a direct extension of RoPE to higher-dimensional domains.

We first derive this form from translation-invariant relative-position attention, then instantiate it with a regular-simplex wave-vector design and analyze its geometric properties.

## 4.1. Fourier-based Generalization of RoPE to $n$ Dimensions

We start from the same assumptions as RoPE: (i) query and key embeddings are position-dependent functions:

$$\mathbf{q}_{\mathbf{x}_1} = f(q, \mathbf{x}_1), \quad \mathbf{k}_{\mathbf{x}_2} = f(k, \mathbf{x}_2), \tag{1}$$

and (ii) the attention kernel depends only on the relative displacement $\mathbf{d} = \mathbf{x}_1 - \mathbf{x}_2$:

$$\langle f(q, \mathbf{x}_1), f(k, \mathbf{x}_2) \rangle = h(q, k, \mathbf{d}). \tag{2}$$

Our goal is to derive a position-dependent embedding form $f(\cdot, \mathbf{x})$ consistent with this translation-invariant relative-position structure. We first view each embedding as an infinite-dimensional Hilbert-space vector with coordinates $\{f_i(q, \mathbf{x})\}_{i=0}^\infty$. The inner product between two such embeddings is

$$\langle f(q, \mathbf{x}_1), f(k, \mathbf{x}_2) \rangle = \sum_{i=0}^\infty f_i(q, \mathbf{x}_1) f_i(k, \mathbf{x}_2)^*. \tag{3}$$

We next specify these coordinates through a function-space representation. We lift the content vector $q$ to a square-integrable function $\gamma(q, \cdot) \in L^2(\mathbb{R}^n)$, and introduce position by translating this function, rather than by assuming a predefined fused content-position form. Given a fixed orthonormal basis $\{\phi_i\}_{i=0}^\infty$ of $L^2(\mathbb{R}^n)$, we define each coordinate as the projection of the translated content function onto a basis element:

$$f_i(q, \mathbf{x}_1) = \langle \gamma(q, \cdot + \mathbf{x}_1), \phi_i(\cdot) \rangle. \tag{4}$$

Here, $q$ selects the underlying content function, $\mathbf{x}_1$ acts through translation, and the fixed basis reads out the coordinates of the resulting Hilbert-space vector. This representation allows us to relate inner products between embeddings to those of the underlying functions. By Parseval's identity, the positional dependence can be transferred from the basis functions to the function $\gamma(\cdot)$ itself, yielding an inner

product between shifted functions:

$$\langle f(q, \mathbf{x}_1), f(k, \mathbf{x}_2) \rangle = \int_{\mathbb{R}^n} \gamma(q, \mathbf{x} + \mathbf{x}_1) \, \gamma(k, \mathbf{x} + \mathbf{x}_2)^* \, d\mathbf{x}$$
$$= \int_{\mathbb{R}^n} \gamma(q, \mathbf{x} + \mathbf{d}) \, \gamma(k, \mathbf{x})^* \, d\mathbf{x}, \tag{5}$$

where the second equality follows from a change of variables $\mathbf{x} \mapsto \mathbf{x} - \mathbf{x}_2$, making the dependence on the relative displacement $\mathbf{d} = \mathbf{x}_1 - \mathbf{x}_2$ explicit. Here, $*$ denotes complex conjugation, ensuring conjugate symmetry of the inner product, and for real-valued $\gamma$, we have $\gamma^* = \gamma$.

To isolate the positional dependence, we fix the content variables $q$ and $k$ and analyze the embedding as a function of position. Let $\Gamma(q, \boldsymbol{\omega})$ denote the Fourier transform of $\gamma(q, \mathbf{x})$. Applying Parseval's theorem to Equation (5) yields:

$$\int_{\mathbb{R}^n} \gamma(q, \mathbf{x} + \mathbf{d}) \, \gamma(k, \mathbf{x})^* \, d\mathbf{x}$$
$$= \int_{\mathbb{R}^n} e^{j\boldsymbol{\omega}^\top \mathbf{d}} \, \Gamma(q, \boldsymbol{\omega}) \, \Gamma(k, \boldsymbol{\omega})^* \, d\boldsymbol{\omega}. \tag{6}$$

The phase factor $e^{j\boldsymbol{\omega}^\top \mathbf{d}}$ captures the relative positional dependence, while the product $\Gamma(q, \boldsymbol{\omega})\Gamma(k, \boldsymbol{\omega})^*$ determines how the content inner product is distributed over frequencies. To ensure that the positional embedding preserves the original content similarity when there is no relative displacement, we impose the initial condition $f(q, 0) = q$ and $f(k, 0) = k$. Setting $\mathbf{x}_1 = \mathbf{x}_2$ gives $\mathbf{d} = 0$, so the relative-position kernel should reduce to the standard content inner product:

$$\langle f(q, \mathbf{x}_1), f(k, \mathbf{x}_1) \rangle = h(q, k, 0)$$
$$= \langle f(q, 0), f(k, 0) \rangle = q^\top k. \tag{7}$$

Under this zero-displacement condition, combining (5), (6), and (7) yields the frequency-domain formulation:

$$\int_{\mathbb{R}^n} \Gamma(q, \boldsymbol{\omega}) \, \Gamma(k, \boldsymbol{\omega})^* \, d\boldsymbol{\omega} = q^\top k. \tag{8}$$

This identity must hold for all content vectors $q$ and $k$. Therefore, the frequency-domain representation must preserve the standard linear kernel after integration over frequencies. A natural way to satisfy this requirement is to represent $\Gamma(q, \boldsymbol{\omega})$ as a frequency-dependent linear projection of $q$. By the Riesz representation theorem (Yosida, 2012), there exists a vector $B(\boldsymbol{\omega}) \in \mathbb{C}^d$ such that

$$\Gamma(q, \boldsymbol{\omega}) = q^\top B(\boldsymbol{\omega}), \quad \int_{\mathbb{R}^n} B(\boldsymbol{\omega}) \, B(\boldsymbol{\omega})^H \, d\boldsymbol{\omega} = I_d, \tag{9}$$

where $H$ denotes the conjugate transpose. We then map the frequency-domain representation back to a position-dependent content function through the inverse Fourier transform:

$$\gamma(q, \mathbf{x}) = \int_{\mathbb{R}^n} \Gamma(q, \boldsymbol{\omega}) \, e^{j\boldsymbol{\omega}^\top \mathbf{x}} \, d\boldsymbol{\omega}. \tag{10}$$

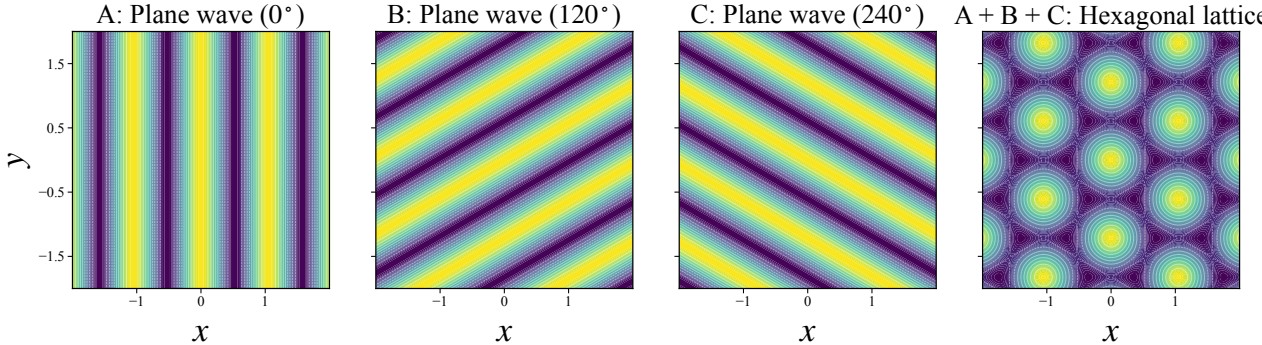

*Figure 3.* Reciprocal-space wave vectors induce real-space positional patterns. In 2D, three wave vectors arranged at $120°$ generate plane waves whose superposition forms a hexagonal lattice.

Substituting the linear form (9) yields

$$\gamma(q, \mathbf{x}) = q^\top \underbrace{\left( \int_{\mathbb{R}^n} B(\omega) \, e^{j\omega^\top \mathbf{x}} \, d\omega \right)}_{:= \phi(\mathbf{x})}. \qquad (11)$$

We thus obtain a factorized representation $\gamma(q, \mathbf{x}) = q^\top \phi(\mathbf{x})$, where $\phi(\mathbf{x})$ serves as a vector-valued Fourier basis. This formulation decouples content and position, interpreting $\gamma(q, \mathbf{x})$ as the projection of $q$ onto a position-dependent basis. To obtain a tractable finite-dimensional embedding, we approximate the continuous Fourier expansion by sampling a finite set of frequencies.

$$\varphi(\mathbf{x}) = \left[ e^{j\omega_1^\top \mathbf{x}}, \ldots, e^{j\omega_M^\top \mathbf{x}} \right]^\top. \qquad (12)$$

With this finite-frequency approximation, the resulting features take the form:

$$f(q, \mathbf{x}) \approx (Wq) \odot \varphi(\mathbf{x}), \qquad (13)$$

where $W$ stacks the frequency-dependent weights $B(\boldsymbol{\omega}_m)^\top$. After separating real and imaginary parts, this construction yields a $2M$-dimensional real embedding. Since the continuous Fourier expansion is approximated by a finite set of frequencies, the embedding inner products provide a finite-frequency approximation of the original kernel. In practice, $W$ can be absorbed into the Transformer query projection matrix $W_Q$, so that positional modulation is applied multiplicatively through $\varphi(\mathbf{x})$.

In the $n$-dimensional setting, both $\boldsymbol{\omega}$ and $\mathbf{x}$ are $n$-dimensional vectors, and a finite embedding is obtained by sampling frequencies from the spectral domain. While the random Fourier features framework samples frequencies i.i.d. from a distribution $p(\boldsymbol{\omega})$ induced by a stationary kernel, this often ties the frequency spectrum to the characteristic scales of the chosen kernel and limits flexibility in modeling multi-scale spatial variations. This limitation becomes more pronounced in high-dimensional settings, where relative displacements can span a wide range of magnitudes. To

address this issue, we next introduce a multi-scale frequency sampling strategy that enables hierarchical coverage of the frequency domain.

### 4.2. Wave Vector Selection

Given the Fourier form derived in Sec. 4.1, the remaining design choice is how to select a finite wave-vector set $\Omega = \{\omega_i\}_{i=1}^M \subset \mathbb{R}^n$. Although these vectors live in reciprocal space, their geometry determines how positional phases resolve directions in real space, and therefore whether the resulting representation is degenerate or directionally biased.

Fig. 3 illustrates this frequency-to-real-space relationship. Each wave vector defines a family of parallel equal-phase hyperplanes, and the superposition of the corresponding plane waves determines the symmetry of the induced positional pattern. In two dimensions, two orthogonal wave vectors induce a square grid tied to axis-aligned phase families, whereas three wave vectors separated by $120°$ form the minimal closed non-axis-aligned configuration and induce a hexagonal lattice. This contrast shows that the number and angular arrangement of wave vectors jointly determine whether the real-space pattern is axis-separable or directionally balanced.

Motivated by this observation, we impose two structural conditions on $\Omega$. First, *coverage* requires the wave vectors to span the ambient space, so that no continuous displacement direction is invisible to the phase system. Second, *symmetry* requires that, under a minimal-redundancy design, the induced real-space phase pattern has as large a symmetry group as possible, rather than remaining separable along coordinate axes. Thus, we seek a compact wave-vector configuration that is rich enough to avoid degeneracy, yet structured enough to produce a coherent and highly symmetric real-space interference pattern.

**Coverage.** We first require the wave-vector set to provide non-degenerate spatial coverage. Each wave vector $\omega_i$

induces a periodic phase constraint

$$\omega_i^\top x = 2\pi k_i, \qquad k_i \in \mathbb{Z}, \tag{14}$$

which corresponds to a family of parallel equal-phase hyperplanes in real space. Stacking the wave vectors row-wise gives

$$\Omega x = 2\pi k, \tag{15}$$

where $\Omega \in \mathbb{R}^{M \times n}$. If $\mathrm{rank}(\Omega) < n$, there exists a nonzero direction $v$ such that $\Omega v = 0$, so the phase system cannot distinguish $x$ from $x + tv$ for any $t \in \mathbb{R}$. Thus, non-degenerate coverage requires

$$\mathrm{rank}(\Omega) = n. \tag{16}$$

This condition guarantees full coverage, but not the symmetry of the induced real-space phase pattern.

**Maximum Symmetry.** Indeed, an axis-wise construction with $M = n$ orthogonal wave vectors already satisfies the rank condition. With equal norms, it can even satisfy the second-order balance condition:

$$\sum_{i=1}^{n} \omega_i \omega_i^\top \propto I_n. \tag{17}$$

Nevertheless, this construction remains coordinate-separable: each Fourier mode is tied to one coordinate direction, producing an orthogonal phase grid in real space. Thus, full rank and second-order balance are not sufficient to remove axis-aligned inductive bias.

To obtain a non-axis-aligned design with minimal additional redundancy, we move to the next smallest choice, $M = n + 1$. This allows the wave vectors to be centered and treated equivalently, without selecting a preferred coordinate axis or preferred pair of directions. The maximally symmetric configuration under this minimal redundancy is the centered regular simplex, whose vectors satisfy

$$\sum_{i=1}^{n+1} \omega_i = 0, \qquad \|\omega_i\| = r, \qquad \langle \omega_i, \omega_j \rangle = -\frac{r^2}{n}, \quad i \neq j. \tag{18}$$

The zero-centroid condition removes a net directional bias, while the equal pairwise inner products ensure that all wavevector directions are related by the same geometry. These identities imply

$$\sum_{i=1}^{n+1} \omega_i \omega_i^\top = \frac{n+1}{n} r^2 I_n, \tag{19}$$

so the aggregate second-order directional energy is identical for every spatial direction. Thus, the regular simplex provides a compact, non-axis-aligned, and maximally symmetric wave-vector set for nD-RoPE.

Using more than $n+1$ directions can further densify angular coverage, but it also introduces additional redundancy and more complex interference among plane waves. Since our goal is a deterministic minimal-redundancy construction, we adopt the regular simplex as the canonical single-scale wave-vector design. In one dimension, the simplex consists of two opposite directions, $\{-r, +r\}$. For real-valued RoPE features, these conjugate Fourier modes are represented by a single positive frequency. Thus, the one-dimensional case reduces to standard RoPE.

### 4.3. nD-RoPE Construction

Having established the criteria of coverage and symmetry, we now summarize the final form of nD-RoPE. Accordingly, the nD-RoPE embedding applied to a query vector $q$ at position $\mathbf{x}$ is defined as

$$f(q, \mathbf{x}) = q \odot \Big[ \underbrace{e^{j(\boldsymbol{\omega}_1^{(1)})^\top \mathbf{x}}, \ldots, e^{j(\boldsymbol{\omega}_M^{(1)})^\top \mathbf{x}}}_{z^{(1)}(\mathbf{x})}$$
$$\| \cdots \| \underbrace{e^{j(\boldsymbol{\omega}_1^{(S)})^\top \mathbf{x}}, \ldots, e^{j(\boldsymbol{\omega}_M^{(S)})^\top \mathbf{x}}}_{z^{(S)}(\mathbf{x})} \Big]^\top. \tag{20}$$

where $S$ is the number of scales, $M$ the number of wave vectors per scale, and $\boldsymbol{\omega}_i^{(s)}$ the $i$-th wave vector at scale $s$. For $n \geq 2$, here $n$ denotes the spatial dimensionality, each scale employs $M = n + 1$ wave vectors arranged as the vertices of a regular simplex in reciprocal space. For $n = 1$, this reduces to the standard 1D RoPE with a single positive frequency. The explicit construction of these vectors is provided in Appendix A. This simplex-based organization induces an isotropic and periodic spatial representation.

Although written in complex form, nD-RoPE is implemented entirely with real-valued block rotations, exactly as in standard RoPE: each phase $e^{j\boldsymbol{\omega}^\top \mathbf{x}}$ corresponds to the cosine–sine pair $\big(\cos(\boldsymbol{\omega}^\top \mathbf{x}), \sin(\boldsymbol{\omega}^\top \mathbf{x})\big)$. Thus, nD-RoPE preserves the computational form of RoPE, replacing only the one-dimensional phase $\omega x$ with the multidimensional phase $\boldsymbol{\omega}^\top \mathbf{x}$ without modifying the attention mechanism. Consequently, existing RoPE implementations and extrapolation techniques based on frequency rescaling or interpolation can be directly adapted to the $n$-dimensional setting. For completeness, we summarize the construction and implementation pipeline in Appendix B.

## 5. Experiments

We empirically evaluate nD-RoPE by addressing the following four research questions. Specifically, we ask: (RQ1) Does nD-RoPE maintain or improve performance on standard benchmarks compared to existing RoPE variants? (RQ2) Does the proposed simplex-based frequency construction reduce directional bias and improve robustness to

*Table 1.* Full ImageNet-1K resolution extrapolation results (Top-1 accuracy,%). All models are trained at $224 \times 224$. For each resolution, the best result *without* YaRN is underlined, and the best result *with* YaRN is highlighted in bold. The training resolution $224 \times 224$ is highlighted with a shaded background to denote the in-domain setting.

| Method | 160 | 192 | 224 | 256 | 320 | 384 | 448 | 512 | 640 | 768 | 896 | 1024 |
|---|---|---|---|---|---|---|---|---|---|---|---|---|
| Learnable PE (DeiT-S) | 76.06 | 79.06 | 80.51 | 80.92 | 80.65 | 79.39 | 77.62 | 75.38 | 70.35 | 64.50 | 57.50 | 50.99 |
| FoPE | 75.63 | 78.47 | 79.91 | 79.79 | 78.64 | 76.84 | 74.31 | 71.69 | 64.94 | 56.08 | 44.16 | 32.62 |
| RoPE-Axial | 76.31 | 79.20 | 80.89 | 81.66 | 81.46 | 80.03 | 78.25 | 76.10 | 67.85 | 53.87 | 35.91 | 20.64 |
| RoPE-Axial + APE | 76.40 | 79.21 | 80.66 | 81.60 | 81.94 | 81.15 | 78.83 | 75.26 | 63.58 | 45.05 | 26.87 | 14.84 |
| RoPE-Mixed | 76.61 | 79.59 | 80.90 | 81.82 | 82.24 | 81.82 | 80.89 | 79.11 | 71.60 | 54.72 | 32.52 | 16.63 |
| RoPE-Mixed + APE | 76.73 | 79.49 | 80.92 | 81.84 | 82.06 | 81.75 | 80.52 | 78.51 | 70.86 | 54.75 | 32.12 | 15.34 |
| nD-RoPE | 77.09 | 79.46 | 81.07 | 81.54 | 82.03 | 81.62 | 80.78 | 79.46 | 74.68 | 66.06 | 52.26 | 35.51 |
| FoPE + YaRN | 75.89 | 78.52 | 79.91 | 80.32 | 79.93 | 78.81 | 77.30 | 75.33 | 70.83 | 65.43 | 59.37 | 53.46 |
| RoPE-Axial + YaRN | 76.30 | 79.21 | 80.88 | 81.70 | 81.80 | 81.38 | 79.72 | 78.48 | 74.97 | 68.34 | 59.13 | 48.02 |
| RoPE-Axial + APE + YaRN | 76.41 | 79.21 | 80.67 | 81.62 | 82.11 | 81.85 | 80.70 | 79.25 | 73.86 | 64.40 | 51.04 | 38.69 |
| RoPE-Mixed + YaRN | 76.62 | **79.59** | 80.90 | 81.85 | **82.41** | **82.17** | **81.78** | 80.93 | 77.34 | 70.01 | 57.99 | 43.48 |
| RoPE-Mixed + APE + YaRN | 76.72 | 79.49 | 80.92 | **81.84** | 82.22 | 82.15 | 81.52 | 80.44 | 76.92 | 70.70 | 61.16 | 49.40 |
| nD-RoPE + YaRN | **77.09** | 79.46 | **81.07** | 81.65 | 82.15 | 82.07 | 81.77 | **81.34** | **79.69** | **77.22** | **73.56** | **68.46** |

*Table 2.* TimeSformer resolution extrapolation results on Kinetics-400 (Top-1 accuracy, %). All models are trained at $224 \times 224$. For each resolution, the best result *without* YaRN is underlined, and the best result *with* YaRN is highlighted in bold. The training resolution $224 \times 224$ is highlighted with a shaded background to indicate the in-domain setting.

| Method | 160 | 192 | 224 | 256 | 320 | 384 | 448 | 512 | 640 | 768 | 896 | 1024 |
|---|---|---|---|---|---|---|---|---|---|---|---|---|
| Learnable PE | 70.05 | 73.86 | 75.61 | 76.42 | 75.92 | 74.68 | 73.39 | 71.45 | 67.57 | 63.51 | 58.83 | 54.23 |
| FoPE | 69.93 | 73.55 | 75.10 | 75.31 | 73.94 | 71.10 | 67.84 | 64.94 | 57.19 | 49.30 | 41.00 | 33.40 |
| RoPE-Axial | 66.25 | 70.53 | 73.23 | 73.92 | 73.32 | 70.94 | 67.50 | 62.57 | 50.53 | 37.29 | 25.21 | 16.13 |
| RoPE-Axial + APE | 66.24 | 71.21 | 73.77 | 74.32 | 73.34 | 70.56 | 66.72 | 61.22 | 46.99 | 32.72 | 20.57 | 12.96 |
| RoPE-Mixed | 66.76 | 70.67 | 73.12 | 73.79 | 73.48 | 71.75 | 68.89 | 65.36 | 55.03 | 42.47 | 29.68 | 18.47 |
| RoPE-Mixed + APE | 66.47 | 70.94 | 73.49 | 74.05 | 73.49 | 71.84 | 69.50 | 65.73 | 56.07 | 43.54 | 30.38 | 19.14 |
| nD-RoPE | 70.63 | 74.43 | 75.85 | 76.09 | 75.44 | 73.89 | 71.17 | 67.48 | 58.92 | 49.17 | 39.01 | 29.21 |
| FoPE + YaRN | 69.93 | 73.55 | 75.10 | 75.78 | 75.67 | 75.63 | 74.41 | 72.72 | 69.39 | 65.72 | 61.07 | 55.87 |
| RoPE-Axial + YaRN | 66.25 | 70.53 | 75.03 | 75.83 | 75.70 | 74.95 | 73.73 | 71.85 | 68.90 | 65.54 | 61.66 | 57.94 |
| RoPE-Axial + APE + YaRN | 66.24 | 71.21 | 74.64 | 75.70 | 75.62 | 74.84 | 73.97 | 72.83 | 69.49 | 65.77 | 62.33 | 58.38 |
| RoPE-Mixed + YaRN | 66.76 | 70.67 | 75.23 | 75.54 | 75.83 | 74.74 | 73.27 | 70.71 | 65.38 | 59.07 | 51.29 | 42.79 |
| RoPE-Mixed + APE + YaRN | 66.47 | 70.94 | 74.51 | 75.72 | 75.49 | 75.00 | 73.97 | 71.92 | 67.58 | 61.69 | 53.52 | 44.16 |
| **nD-RoPE + YaRN** | **70.63** | **74.43** | **75.85** | **76.50** | **76.27** | **75.88** | **74.22** | **72.91** | **69.78** | **66.49** | **63.19** | **59.23** |

rotation? (RQ3) Can nD-RoPE generalize better under scale shifts? (RQ4) How do the key design choices of nD-RoPE influence performance?

**Benchmark.** To answer these questions, we evaluate nD-RoPE across a diverse set of vision and geometric tasks with varying input dimensionalities: (i) 2D image classification on ImageNet-1K (Deng et al., 2009) using ViT-S (Dosovitskiy et al., 2021); (ii) video recognition on Kinetics-400 (Kay et al., 2017) with joint spatial–temporal modeling based on TimeSformer (Bertasius et al., 2021); (iii) 3D point cloud segmentation on ModelNet40 (Wu et al., 2015) using Point Transformer (Zhao et al., 2021); (iv) large-scale outdoor point cloud segmentation on SemanticKITTI (Behley et al., 2019) with Point Transformer v2 (Wu et al., 2022).

**Comparison Methods.** To contextualize the effectiveness of nD-RoPE, we compare it with representative positional embedding approaches, including (1) learnable absolute positional embeddings (APE) (Devlin et al., 2019); (2) axial

RoPE variants (Su et al., 2024; Ma et al., 2025), which apply independent rotary embeddings along each axis; and (3) mixed RoPE variants (Heo et al., 2024; Ostmeier et al., 2025), which introduce cross-dimensional interactions via learnable frequency mixing. We also consider hybrid variants that combine relative and absolute embeddings, namely (4) Axial RoPE + APE and (5) Mixed RoPE + APE. For image and video tasks involving resolution extrapolation, we further include FoPE (Hua et al., 2025) as an extrapolation baseline, and integrate YaRN (Peng et al., 2023) into all rotary-based methods to ensure a fair comparison under extended resolution settings.

**Experimental Setup.** Unless otherwise specified, we adopt the official configurations of each backbone model as the reference setting. For different positional embedding variants, we use the same backbone architecture and matched training protocol. This design keeps the comparison focused on the effect of positional embedding. Full implementation details are provided in Appendix C.

*Table 3.* Point cloud density extrapolation results on ModelNet40 using Point Transformer. All models are trained with 2048 input points. Instance-average mIoU (%) is reported, and the in-domain setting is indicated by a shaded background.

| Method | 256 | 768 | 1024 | 1536 | *2048* | 3072 | 4096 |
|---|---|---|---|---|---|---|---|
| Learnable Rel. PE | 43.78 | 73.35 | 77.51 | 81.66 | 82.58 | 80.82 | 78.15 |
| 3D RoPE-Axial | 48.22 | 70.16 | 73.74 | 80.14 | 80.98 | 79.13 | 76.14 |
| 3D RoPE-Axial + APE | 52.71 | 75.12 | 77.94 | 81.42 | 81.92 | 80.48 | 78.09 |
| 3D RoPE-Mixed | 40.41 | 73.23 | 76.48 | 80.91 | 81.40 | 80.15 | 77.86 |
| 3D RoPE-Mixed + APE | 45.25 | 68.95 | 72.52 | 79.98 | 81.13 | 78.86 | 75.25 |
| nD-RoPE (vector attention) | **55.37** | **78.90** | **82.65** | **85.76** | **85.97** | **84.92** | **82.75** |
| nD-RoPE (standard dot-product) | 46.13 | 76.07 | 80.85 | 84.66 | 85.07 | 83.80 | 81.06 |

*Table 4.* SemanticKITTI point cloud segmentation results using Point Transformer v2. Models are trained with a grid size of 0.05. We report instance-average mIoU (%) under varying grid resolutions, and the in-domain setting is indicated by a shaded background.

| Method | 0.02 | 0.03 | 0.04 | *0.05* | 0.075 | 0.10 | 0.15 |
|---|---|---|---|---|---|---|---|
| Learnable Rel. PE | 72.44 | 72.51 | 72.73 | 71.70 | 68.46 | 64.68 | 50.20 |
| 3D RoPE-Axial | 68.78 | 68.73 | 69.61 | 70.25 | 68.92 | 64.12 | 51.76 |
| 3D RoPE-Axial + APE | 67.02 | 67.68 | 69.15 | 69.82 | 68.51 | **65.57** | **54.19** |
| 3D RoPE-Mixed | 67.54 | 67.74 | 68.09 | 68.40 | 67.66 | 63.04 | 49.57 |
| 3D RoPE-Mixed + APE | 69.10 | 69.25 | 70.21 | 70.28 | 68.88 | 61.95 | 52.07 |
| nD-RoPE (vector attention) | **72.70** | **72.91** | **73.14** | **71.91** | **69.84** | 64.37 | 53.53 |
| nD-RoPE (standard dot-product) | 66.90 | 66.71 | 67.46 | 67.12 | 65.25 | 57.87 | 50.06 |

## 5.1. In-Domain Performance on Standard Benchmarks

We first evaluate the in-domain performance of nD-RoPE, where the input resolution, point density, or grid size matches the training configuration. These settings are highlighted by gray shading in Tables 1–4.

On 3D benchmarks, nD-RoPE achieves consistent in-domain gains under identical training conditions. On ModelNet40 (Table 3), nD-RoPE with vector attention reaches 85.97% mIoU at 2048 points, outperforming all axial and mixed RoPE baselines. It remains competitive with standard dot-product attention, suggesting that the simplex-based frequency construction provides a strong inductive bias independent of the attention formulation. On SemanticKITTI (Table 4), nD-RoPE also attains the best in-domain mIoU at the training grid size of 0.05, showing that this advantage persists in large-scale sparse scenes without relying on axis-wise decomposition or learnable positional parameters.

This trend extends to 2D and spatiotemporal settings. On ImageNet-1K (Table 1), nD-RoPE achieves 81.07% top-1 accuracy at $224 \times 224$, exceeding existing RoPE variants with fixed frequencies. On Kinetics-400 (Table 2), it yields the strongest in-domain result among non-YaRN methods.

Overall, nD-RoPE consistently improves or preserves in-domain performance across modalities, while incurring only negligible to modest computational overhead, as detailed in Appendix D.4.

## 5.2. Geometric Isotropy and Rotational Robustness

To evaluate geometric isotropy, we perform a zero-shot rotational robustness test by rotating ImageNet validation images while keeping all models fixed. Each image is resized to $256 \times 256$, rotated by a given angle, and center-cropped to $224 \times 224$, ensuring that rotations genuinely disrupt axis-aligned spatial structure.

As shown in Table 5, nD-RoPE exhibits stronger rotational robustness than axial and mixed RoPE variants. While baseline methods achieve their best performance at $0°$ and degrade rapidly as the rotation angle increases, nD-RoPE maintains higher accuracy across the sampled non-zero angles, with the gap becoming especially pronounced at intermediate rotations such as $30°$, $120°$, and $150°$. Interestingly, we also observe a partial performance recovery for all methods at $180°$, where the rotation restores alignment with the original coordinate axes. Despite this recovery, nD-RoPE continues to outperform the baselines, highlighting its reduced reliance on privileged directional information.

This behavior stems from the regular simplex frequency construction in nD-RoPE, which provides isotropic coverage of the frequency space. In contrast, axis-wise and mixed RoPE variants rely on a small set of privileged directions, making them sensitive to rotations that mix spatial axes.

## 5.3. Resolution and Density Extrapolation

Extrapolation beyond the training scale is a revealing stress test for positional embedding schemes, as it probes whether

*Table 5.* Zero-shot rotational robustness on ImageNet-1K (Top-1 accuracy, %). Each model is evaluated under fixed rotations without fine-tuning. The best result at each rotation angle is highlighted in bold.

| Angle | DeiT (APE) | RoPE-Axial | RoPE-Axial + APE | RoPE-Mixed | RoPE-Mixed + APE | Ours (nD-RoPE) |
|---|---|---|---|---|---|---|
| $0°$ | 80.44 | 81.00 | 80.94 | 80.99 | **81.13** | 80.81 |
| $30°$ | 70.12 | 70.61 | 70.89 | 71.34 | 71.12 | **78.51** |
| $60°$ | 56.37 | 58.12 | 57.05 | 59.08 | 58.31 | **60.83** |
| $90°$ | 56.01 | 56.91 | 56.52 | 57.10 | 56.91 | **60.50** |
| $120°$ | 44.74 | 45.78 | 46.20 | 47.29 | 47.30 | **55.92** |
| $150°$ | 45.99 | 48.10 | 47.15 | 48.55 | 48.99 | **59.28** |
| $180°$ | 58.29 | 58.85 | 58.52 | 59.31 | 59.28 | **62.53** |

a model encodes geometric relationships that generalize across changes in resolution and sampling density (Ding et al., 2024; Zhao et al., 2024). Prior work has shown that ViTs with absolute positional embeddings often rely on interpolation heuristics and degrade outside the training domain, whereas rotary-based methods offer a more principled mechanism for extrapolation through continuous relative position modeling (Su et al., 2024; Heo et al., 2024; Peng et al., 2023).

We therefore evaluate nD-RoPE under resolution extrapolation for images and videos, and under density extrapolation for point clouds, as summarized in Tables 1–4. For images and videos, extrapolation corresponds to increasing the spatial resolution beyond the training scale, which alters relative token distances. For point clouds, varying the number of input points or voxel resolution changes local sampling density while preserving the global spatial extent. For intuitive illustration, we further provide qualitative visualizations of extrapolation behavior in Appendix D.2.

Across all settings, nD-RoPE exhibits consistently slower performance degradation than axis-wise and mixed RoPE variants as the evaluation scale departs from the training configuration (Tables 1, 2, 3, and 4). This robustness stems from the regular simplex frequency construction, which provides isotropic coverage of the frequency space and avoids privileging specific coordinate axes, leading to relative positional representations that remain stable under scale shifts. RoPE-style embeddings can be further enhanced by extrapolation techniques such as YaRN (Peng et al., 2023) and FoPE (Hua et al., 2025), which modify or correct frequency behavior under scale shifts. Since nD-RoPE preserves the core RoPE phase structure $e^{j\omega^\top x}$, it is naturally compatible with such techniques. The results show that these methods provide additional extrapolation gains, while nD-RoPE retains its relative advantage, indicating that simplex-based wave-vector geometry is complementary to frequency-scaling or frequency-correction methods.

In point cloud tasks, this advantage is further amplified when nD-RoPE is integrated with vector attention architectures. As shown in Tables 3 and 4, nD-RoPE with vector attention consistently outperforms its dot-product counterpart under sparse sampling and coarse voxelization, indicating that applying isotropic rotary modulation directly to relative geometric relations aligns naturally with the relation-centric design of Point Transformers.

Overall, these results demonstrate that nD-RoPE provides a stronger geometry-aligned inductive bias for handling scale shifts across images, videos, and point clouds.

### 5.4. Ablation Study

We conduct ablation studies to examine how attention heads, multi-scale frequency allocation, and frequency design affect nD-RoPE performance. Under a fixed embedding budget, increasing the number of attention heads reduces the number of frequency scales assigned to each head, while concentrating too many scales in fewer heads limits attention diversity. We observe that performance peaks when frequency scales are evenly distributed across heads, whereas extreme allocations consistently degrade performance. We further analyze the frequency scale ratio and derive a theoretical upper bound on the RoPE frequency base to avoid redundancy and phase ambiguity in positional encoding. We then empirically validate this analysis by studying the effect of the frequency base $\theta$ on the Point Transformer benchmark, and show that the experimental results closely match the theoretical predictions. Detailed results of the above ablation studies are provided in Appendix D.3.

## 6. Conclusion

We presented nD-RoPE, a principled generalization of RoPE to arbitrary-dimensional Euclidean spaces. By treating positions and frequencies as unified $n$-dimensional vectors within a Fourier-based, translation-invariant framework, nD-RoPE provides a decomposition-free formulation beyond axis-wise constructions. Its multi-scale regular-simplex wave-vector design ensures non-degenerate, directionally balanced frequency coverage. Experiments across images, videos, and point clouds demonstrate improved in-domain performance, rotational robustness, and scale extrapolation, highlighting the importance of wave-vector geometry for high-dimensional positional embedding.

## Acknowledgements

This work was supported in part by the U.S. National Science Foundation under Grant No. 2343646.

## Impact Statement

This work advances positional embedding for high-dimensional representations. Its societal impact depends on downstream applications, and we do not identify specific risks beyond those generally associated with machine learning systems.

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

## A. Derivation of Wave Vectors Based on Regular Simplex

This section constructs a regular simplex in an $n$-dimensional hyperplane starting from the standard orthogonal basis in $\mathbb{R}^{n+1}$. We then derive a dimensionality reduction procedure for the associated wave vectors while preserving their geometric structure. The resulting wave vectors are used to provide isotropic frequency directions.

In $\mathbb{R}^{n+1}$, the standard orthogonal basis consists of vectors $\mathbf{e}_1, \mathbf{e}_2, \ldots, \mathbf{e}_{n+1}$, where each basis vector $\mathbf{e}_i$ is defined as

$$\mathbf{e}_i = (0, 0, \ldots, 1, \ldots, 0), \tag{21}$$

with the 1 appearing in the $i$-th coordinate and 0 elsewhere. These vectors form $n+1$ affinely independent points in $\mathbb{R}^{n+1}$, defining the vertices of a regular simplex in the affine hyperplane $\sum_{j=1}^{n+1} x_j = 1$.

However, this simplex is not centered at the origin and lies in an affine hyperplane rather than a linear subspace. To address this, we translate the standard basis vectors so that their centroid lies at the origin. Let

$$\boldsymbol{\omega}_i = \mathbf{e}_i - \frac{1}{n+1}\mathbf{L}, \qquad \mathbf{L} = (1, 1, \ldots, 1) \in \mathbb{R}^{n+1}. \tag{22}$$

By construction, all $\boldsymbol{\omega}_i$ satisfy

$$\sum_{j=1}^{n+1} (\boldsymbol{\omega}_i)_j = 0, \tag{23}$$

and thus lie in the same $n$-dimensional hyperplane $\sum_{j=1}^{n+1} x_j = 0$. Furthermore, a direct computation shows that

$$\langle \boldsymbol{\omega}_i, \boldsymbol{\omega}_j \rangle = \begin{cases} \frac{n}{n+1}, & i = j, \\ -\frac{1}{n+1}, & i \neq j, \end{cases} \tag{24}$$

which implies that all pairwise distances are equal. Therefore, $\{\boldsymbol{\omega}_i\}_{i=1}^{n+1}$ form the vertices of a regular simplex centered at the origin. Stacking the wave vectors $\boldsymbol{\omega}_i^\top$ as rows yields the wave vector matrix

$$\boldsymbol{\Omega}^{(n+1)} \in \mathbb{R}^{(n+1) \times (n+1)}. \tag{25}$$

Since these vectors lie in the $n$-dimensional hyperplane $\sum_{j=1}^{n+1} x_j = 0$ and span this hyperplane, the matrix $\boldsymbol{\Omega}^{(n+1)}$, whose rows are the wave vectors $\boldsymbol{\omega}_i^\top$, has rank $n$. To obtain an explicit $n$-dimensional representation while preserving the geometric relations among the wave vectors, we compute the singular value decomposition

$$\boldsymbol{\Omega}^{(n+1)} = \mathbf{U}\boldsymbol{\Sigma}\mathbf{W}^\top, \tag{26}$$

where $\boldsymbol{\Sigma}$ contains exactly $n$ nonzero singular values. Let

$$\mathbf{W}^{(n)} = [\mathbf{w}_1, \ldots, \mathbf{w}_n] \tag{27}$$

denote the right singular vectors corresponding to the nonzero singular values. These vectors form an orthonormal basis for the row space of $\boldsymbol{\Omega}^{(n+1)}$, which is the $n$-dimensional simplex hyperplane. We then define the reduced wave-vector matrix by

$$\boldsymbol{\Omega}^{(n)} = \boldsymbol{\Omega}^{(n+1)}\mathbf{W}^{(n)} \in \mathbb{R}^{(n+1) \times n}. \tag{28}$$

This projection preserves pairwise inner products, since

$$\boldsymbol{\Omega}^{(n)}\boldsymbol{\Omega}^{(n)\top} = \boldsymbol{\Omega}^{(n+1)}\mathbf{W}^{(n)}\mathbf{W}^{(n)\top}\boldsymbol{\Omega}^{(n+1)\top} = \boldsymbol{\Omega}^{(n+1)}\boldsymbol{\Omega}^{(n+1)\top}. \tag{29}$$

Therefore, the reduced matrix $\boldsymbol{\Omega}^{(n)}$ preserves the relative magnitudes and angular relationships of the original simplex wave vectors, providing an explicit $n$-dimensional realization suitable for constructing isotropic and stable positional encodings.

## B. Implementation Pipeline of nD-RoPE

Fig. 4 provides an overview of the nD-RoPE implementation pipeline. The input positions $X$ are the spatial or spatiotemporal coordinates of tokens, such as image patch coordinates, video patch coordinates, or point-cloud coordinates. The wave vectors $\Omega$ are generated from the regular-simplex construction and optionally rotated independently for each attention head. For each token position $x$ and wave vector $\omega$, nD-RoPE computes the phase $\omega^\top x$ and applies the corresponding complex rotation $e^{j\omega^\top x}$ to the query and key features. Algorithm 1 summarizes the full procedure and shows that nD-RoPE can be incorporated into standard attention mechanisms without modifying the attention computation.

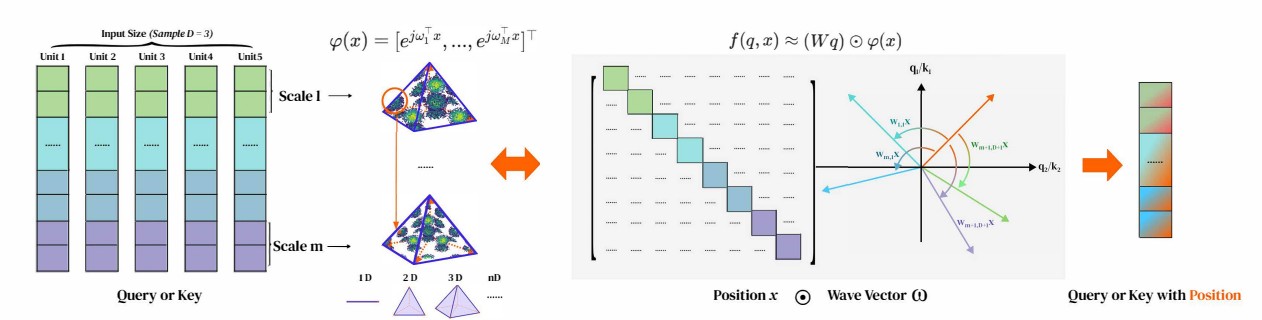

*Figure 4.* Visualization of the nD-RoPE implementation pipeline. Token coordinates $x$ are projected onto multi-scale regular-simplex wave vectors $\omega$, producing phases $\omega^\top x$. These phases generate rotary factors $e^{j\omega^\top x}$, which are applied to query and key features while leaving the attention mechanism unchanged.

## C. Implementation Details

All experiments were conducted using four NVIDIA A100 GPUs with $40\,\text{GB}$ memory, running CUDA 12.1. We use AdamW as the optimizer, and adjust the learning rate, weight decay, and per-GPU batch size according to the available training environment. All other hyperparameters and architectural settings are kept identical to those in the original training setup to ensure fair comparison. For reproducibility and ease of reference, we summarize the training settings in this appendix.

**ImageNet-1k Image Classification with ViT-S.** We adopt a DeiT-style AdamW training recipe for all ImageNet-1k experiments to ensure a unified and fair comparison. All models are based on the ViT-S / DeiT-S architecture with a patch size of $16 \times 16$ and an input resolution of $224 \times 224$. Models are trained for 400 epochs using the AdamW optimizer with a base learning rate of $5 \times 10^{-4}$ and a weight decay of $0.05$. A cosine learning rate schedule with a linear warmup of 5 epochs is employed. Label smoothing with a factor of $0.1$ is applied during training.

Standard data augmentation strategies follow the original rope-vit(Heo et al., 2024) training protocol, including Mixup, CutMix, color jittering, and repeated augmentation. Additional regularization techniques such as stochastic depth with a drop-path rate of $0.1$ are adopted following the standard DeiT-style AdamW training recipe. All other architectural settings and training hyperparameters follow the standard DeiT training recipe unless otherwise specified. We report single-crop top-1 accuracy on the ImageNet-1k validation set.

For resolution extrapolation beyond the training scale, we additionally integrate YaRN (Peng et al., 2023) into all RoPE-based methods, including nD-RoPE. YaRN hyperparameters are selected via grid search. Specifically, we apply a resolution-dependent scaling strategy in which the scaling factor increases linearly with the input resolution, using fixed hyperparameters `cutoff` $= 0.6$, `sharpness` $= 8.0$, and `power` $= 1.0$. This allows a fair comparison of extrapolation behavior under a shared scaling mechanism.

**Kinetics-400 Video Recognition with Timesformer.** We evaluate nD-RoPE on the Kinetics-400 video classification benchmark using a ViT-Base backbone with a patch size of $16 \times 16$ and the divided space-time attention mechanism. Unless otherwise specified, we follow the standard TimeSformer training protocol for video modeling, including the number of input frames, temporal sampling rate, spatial cropping strategy, and attention structure. Specifically, all models are trained using 8 input frames sampled with a temporal stride of 32, with spatial jittering scales of $[256, 320]$ and a crop size of 224. The backbone architecture, input resolution, and attention formulation are kept identical across different positional embedding variants to ensure a fair comparison.

For optimization, we adopt a modern AdamW-based training recipe. All models are fine-tuned for 15 epochs starting from a ViT-B/16 pretrained model, using the AdamW optimizer with a base learning rate of $1 \times 10^{-4}$, a cosine learning rate schedule, and a linear warmup of 3 epochs. The pretrained ViT-B/16 weights are obtained from publicly available sources. Weight decay is set to $0.05$. Additional regularization techniques, including stochastic depth with a drop-path rate of $0.2$ and Mixup/CutMix augmentation, are applied following common practices in recent ViT-based video models. We report top-1 classification accuracy using single-clip evaluation with three spatial crops on the Kinetics-400 validation set.

For spatial resolution extrapolation, we incorporate YaRN (Peng et al., 2023) into all RoPE-based variants. YaRN

---

**Algorithm 1** nD-RoPE with Regular Simplex Wave Vectors

---

**Require:** Positions $X \in \mathbb{R}^{N \times d}$, number of heads $H$, per-head dimension $D$ (must be a multiple of $2(d+1)$), base $\theta$
**Ensure:** Rotary phases $\Phi \in \mathbb{C}^{H \times N \times (D/2)}$
  1: **(1) Regular simplex construction**
  2: Construct centered simplex vertices $P = I_{d+1} - \frac{1}{d+1}\mathbf{1}\mathbf{1}^\top$
  3: Compute $W \in \mathbb{R}^{(d+1) \times d}$ as the right singular vectors of $P$
  4: Project and normalize wave vectors $\Omega = \mathrm{NormalizeRows}(PW)$
  5: **(2) Head-wise wave vectors**
  6: **for** $h = 1, \ldots, H$ **do**
  7:      Sample orthogonal rotation $Q_h \in SO(d)$
  8:      $W_h \leftarrow \Omega Q_h^\top$
  9: **end for**
10: **(3) Multi-scale phase construction**
11: Let $M = d+1$, $S = D/(2M)$
12: Define frequency scales $\alpha_s = \theta^{-s/S}$ for $s = 0, \ldots, S-1$
13: **for** $h = 1, \ldots, H$ **do**
14:      Project positions: $Z_h = XW_h^\top \in \mathbb{R}^{N \times M}$
15:      Compute angles $\psi_h[n, m, s] = Z_h[n, m] \cdot \alpha_s$
16:      Flatten $\psi_h$ to shape $(N, D/2)$
17:      $\Phi_h \leftarrow \exp(i\,\psi_h)$
18: **end for**
19: **(4) RoPE rotation (attention unchanged)**
20: **for** each attention head $h$ **do**
21:      $q_h \leftarrow \text{nD-RoPE}(q_h, \Phi_h)$
22:      $k_h \leftarrow \text{nD-RoPE}(k_h, \Phi_h)$
23: **end for**

---

hyperparameters are selected via grid search. We anchor the scaling at $F_{\max} = 8$ for resolution 1024 and apply a power-law scaling to intermediate resolutions, with `cutoff` $= 0.3$, `sharpness` $= 8.0$, and `power` $= 0.75$. This setup enables a consistent evaluation of extrapolation performance in the spatiotemporal setting.

**ModelNet40 Point Cloud Segmentation with Point Transformer.** We evaluate nD-RoPE on the ModelNet40 point cloud part segmentation task using the Point Transformer architecture. Unless otherwise specified, we follow the standard training protocol of the original Point Transformer, including the network depth, neighborhood size, and embedding dimension. Specifically, the model consists of 4 transformer blocks with an embedding dimension of 512 and a neighborhood size of 16.

For training, we adopt the AdamW optimizer and train all models for 200 epochs. The learning rate and batch size are adjusted according to the training configuration, while other architectural settings are kept identical across different positional embedding variants to ensure a fair comparison.

**SemanticKITTI LiDAR Semantic Segmentation with Point Transformer V2.** We evaluate nD-RoPE on the SemanticKITTI benchmark using the Point Transformer v2 architecture. Unless otherwise specified, we follow the standard training and data processing protocol of the original Point Transformer v2 model. The backbone architecture, including the hierarchical encoder–decoder design, neighborhood size, feature dimensions, and grid-based downsampling strategy, is kept identical across all experiments to ensure a fair comparison.

Specifically, the model adopts four encoder stages with channel dimensions of $\{96, 192, 384, 512\}$ and a neighborhood size of 16 at each stage. The decoder mirrors the encoder structure with corresponding channel dimensions. All models are trained for 50 epochs using the AdamW optimizer with a OneCycle learning rate schedule. The base learning rate is set to $2 \times 10^{-3}$ with a weight decay of 0.01. Stochastic depth is applied with a drop-path rate of 0.2 for regularization.

Standard data augmentation strategies for LiDAR point clouds are employed, including random rotation, scaling, flipping, jittering, grid sampling, and spherical cropping. We use a combined Cross-Entropy and Lovász loss for supervision. Training is conducted with a total batch size of 8 across all GPUs. During inference, we follow the standard multi-view evaluation

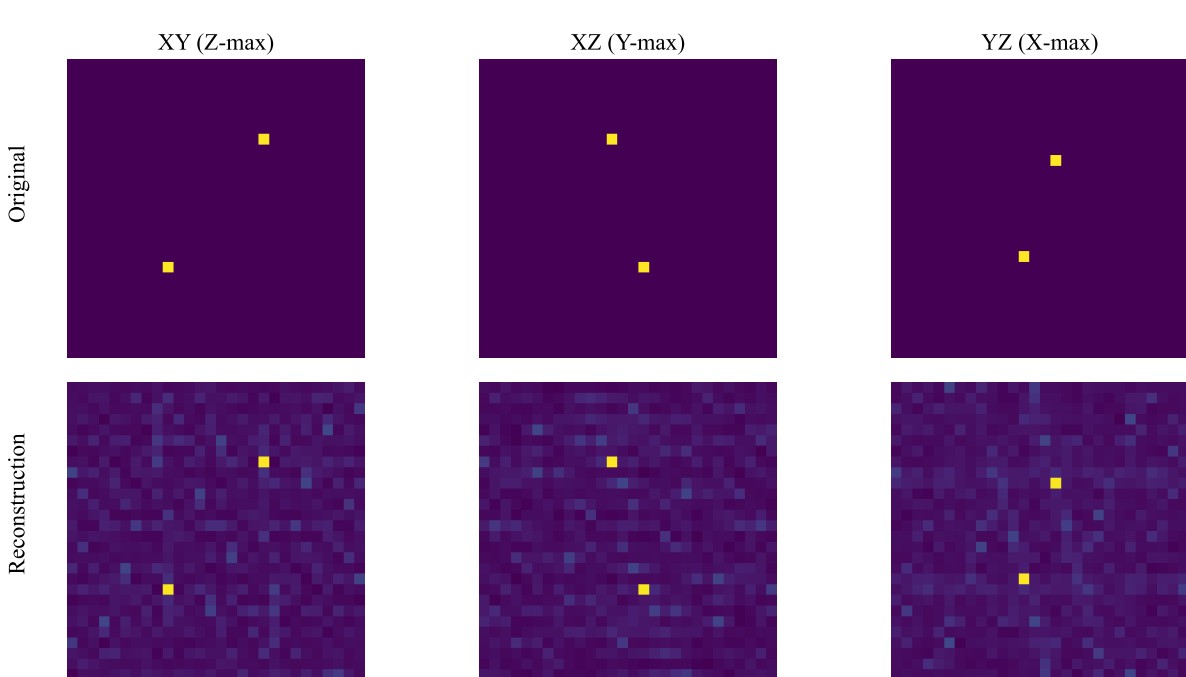

*Figure 5.* **3D NUFT reconstruction with nD-RoPE.** Maximum-intensity projections of the original and reconstructed 3D impulse signals along the XY, XZ, and YZ planes. The reconstruction preserves sharp and isotropic responses across all directions, without axis-aligned artifacts.

protocol with test-time augmentation as in the original setup.

## D. Additional Analysis and Experiments

### D.1. 3D Analysis of nD-RoPE

In this subsection, we provide additional qualitative results for the 3D case of nD-RoPE. We visualize both the NUFT reconstruction behavior and the corresponding frequency distributions to complement the 2D analysis presented in the main text.

As shown in Fig. 5, the NUFT reconstruction behavior observed in the 2D setting extends naturally to three-dimensional signals. Across all orthogonal projections, nD-RoPE is able to recover impulse responses with sharp localization and without introducing axis-aligned artifacts. This indicates that the joint encoding of positional information via $\exp(i\boldsymbol{\omega}^{\top}\boldsymbol{x})$ remains effective in higher dimensions, and that directional coverage is preserved beyond axis-wise bases. And the frequency-domain visualizations in Fig. 6 further support this observation from a geometric perspective. In 3D, the frequency vectors sampled by nD-RoPE form structured concentric shells with approximately uniform angular coverage, while their orthogonal projections exhibit near-circular and symmetric patterns. These properties closely mirror those observed in the 2D analysis and confirm that the isotropic multi-scale design of nD-RoPE generalizes consistently as the spatial dimensionality increases.

### D.2. Qualitative Visualization of Extrapolation

Fig. 7 visualizes the extrapolation behavior of different positional encoding schemes across images, videos, and point clouds. Except for SemanticKITTI with Point Transformer v2, nD-RoPE consistently exhibits slower performance degradation as the evaluation resolution or point density deviates from the training configuration. This trend is most evident in image and video resolution extrapolation as well as ModelNet40 density extrapolation, where axis-wise and mixed RoPE variants rapidly deteriorate under large scale shifts.

These observations align with the theoretical design of nD-RoPE: its isotropic, simplex-based frequency construction

Theoretical Frequency Distribution (nD-RoPE, 3D)

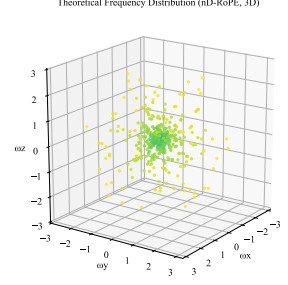

Orthogonal Frequency Projections (nD-RoPE, 3D)

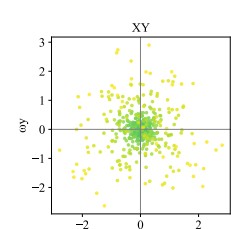 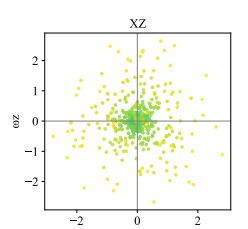 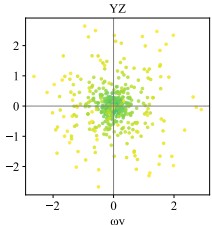

*(a)* 3D frequency distribution (colored by scale)

*(b)* Orthogonal frequency projections (XY / XZ / YZ)

*Figure 6.* **Frequency distribution of nD-RoPE in 3D.** (a) Frequency vectors $\boldsymbol{\omega} \in \mathbb{R}^3$ sampled from multi-scale simplex constructions, forming structured concentric shells in the 3D frequency space. (b) Orthogonal projections onto the XY, XZ, and YZ planes, showing near-circular and symmetric patterns that confirm isotropic multi-scale coverage across all directions.

preserves well-conditioned relative positional representations under changes in resolution or sampling density, whereas axis-aligned encodings privilege specific directions and mixed variants may suffer from anisotropic frequency collapse outside the training regime.

On SemanticKITTI, axial RoPE slightly outperforms nD-RoPE at certain coarse grid resolutions. We attribute this to the strong axis-aligned structure of outdoor LiDAR scenes, where geometric variations often align with sensor axes, favoring axis-wise inductive biases. Nevertheless, nD-RoPE remains competitive across all grid sizes and avoids the severe degradation observed in mixed RoPE variants, indicating robust generalization even in axis-dominated settings.

### D.3. Additional Ablation Studies

We present additional ablation studies to analyze the design choices of nD-RoPE, focusing on (i) the allocation of multi-scale frequencies across attention heads and (ii) the choice of the RoPE frequency base $\theta$.

**Scale–Head Allocation.** All ablation experiments are conducted under a fixed embedding dimension of 384. For 2D inputs, each scale uses three simplex wave vectors, each represented by a cosine–sine pair, resulting in six embedding dimensions per scale. Under this setting, the total number of positional channels remains constant. Only the allocation of scales across attention heads is varied, while all other training and architectural settings are kept identical.

Table 6 shows that neither extremely few nor excessively many scales per head lead to optimal performance. Configurations with balanced scale-to-head allocation (e.g., 6×10 and 4×16) consistently achieve higher Top-1 accuracy across resolutions. In contrast, allocating too many scales to a single head (64×1) or distributing too few scales across many heads (1×64) results in noticeable performance degradation. This trend suggests that nD-RoPE benefits from balancing frequency diversity and attention diversity, as extreme allocations either over-concentrate multi-scale information within a single head or fragment it across insufficient per-head capacity.

*Table 6.* Effect of Scale Count and Head Allocation on Top-1 Accuracy. The best result at each resolution is highlighted in bold.

| Scale × Heads | 160 | 192 | *224* | 256 | 320 | 384 | 448 | 512 |
|---|---|---|---|---|---|---|---|---|
| 64×1 | 67.54 | 69.06 | 70.99 | 71.55 | 72.32 | 73.56 | 76.31 | 77.94 |
| 32×2 | 73.36 | 75.56 | 77.89 | 77.37 | 77.20 | 79.16 | 79.71 | **80.16** |
| 16×4 | 76.87 | 78.45 | 80.17 | 80.40 | 80.92 | 81.10 | 80.20 | 78.90 |
| 6×10 | **77.09** | **79.46** | **81.07** | **81.54** | **82.03** | 81.62 | **80.78** | 79.46 |
| 4×16 | 76.42 | 78.96 | 80.55 | 80.88 | 81.74 | **81.94** | 80.62 | 79.22 |
| 2×32 | 64.18 | 66.50 | 69.57 | 68.64 | 71.64 | 75.01 | 79.16 | 80.24 |
| 1×64 | 58.00 | 63.91 | 65.93 | 64.50 | 67.41 | 69.74 | 73.09 | 74.80 |

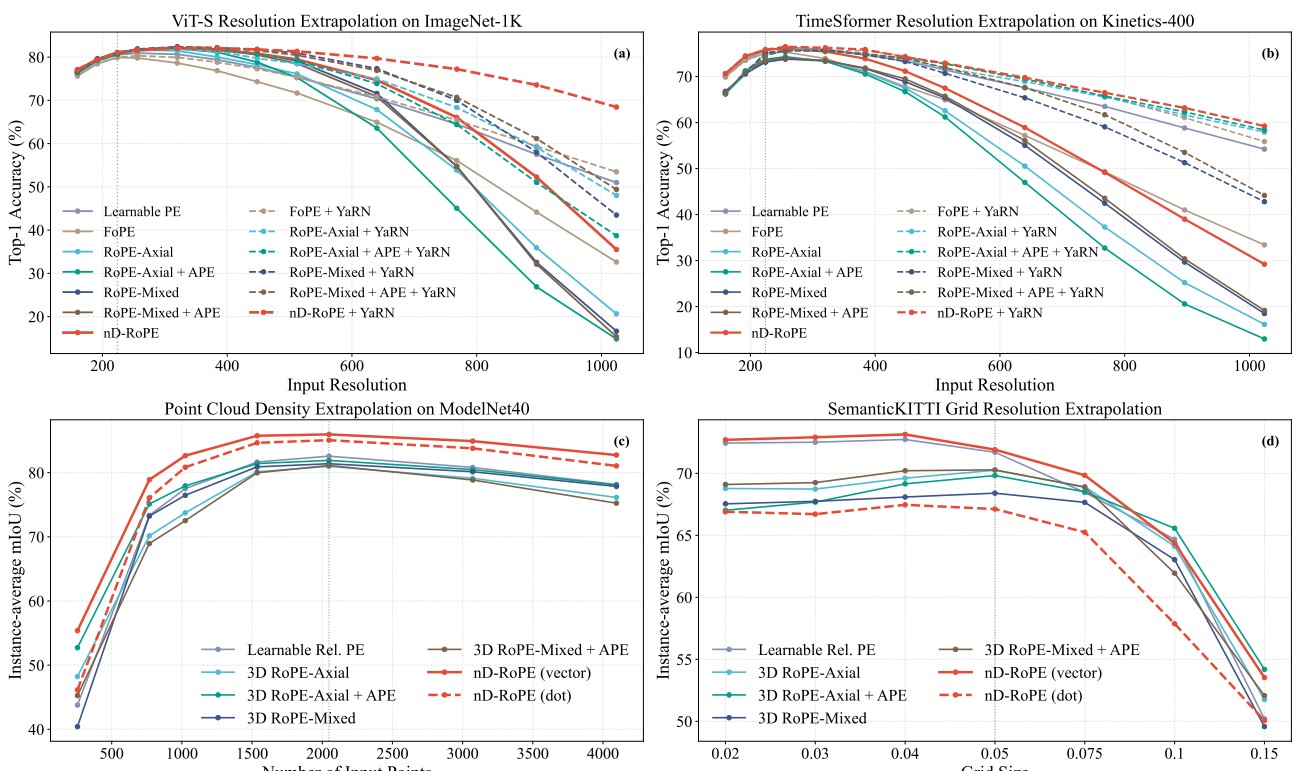

*Figure 7.* Resolution and density extrapolation performance across different modalities. (a) ImageNet-1K image resolution extrapolation with ViT-S. (b) Kinetics-400 video resolution extrapolation with TimeSformer. (c) ModelNet40 point cloud density extrapolation with Point Transformer. (d) SemanticKITTI grid resolution extrapolation with Point Transformer v2.

**Frequency Base.** Table 7 presents an ablation study on the RoPE frequency base $\theta$ for nD-RoPE with vector attention under varying input point densities. Across all settings, we observe that moderate frequency bases (e.g., $\theta = 100$) consistently achieve the best performance, while both smaller bases (e.g., $\theta = 2$) and excessively large bases (e.g., $\theta = 10^4$ or $10^6$) lead to noticeable performance degradation, particularly under sparse and dense input regimes.

This empirical trend aligns well with the theoretical upper-bound analysis in Eq. 37. For the Point Transformer configuration used in this experiment, the embedding dimension is $512$ with $4$ attention heads, yielding a per-head dimension of $D_{\text{head}} = 128$. In the 3D setting, nD-RoPE employs $M = d+1 = 4$ simplex wave vectors per scale. According to Eq. 37, the frequency base is bounded by $\theta \leq \exp\left(D_{\text{head}}/(2Md)\right) = \exp\left(128/(2 \cdot 4 \cdot 3)\right) \approx 207$, indicating that the optimal frequency base should lie on the order of $10^2$. This prediction is consistent with Table 7, where $\theta = 100$ achieves the strongest and most stable performance across all point densities.

### D.4. Computational Complexity Analysis

Table 8 summarizes the computational cost of nD-RoPE across ViT and point cloud Transformer architectures. For image and video Transformers, nD-RoPE only modifies the frequency construction of rotary embeddings without introducing additional attention cost. As a result, the overall FLOPs and parameter counts remain nearly identical to the corresponding baselines, with only a marginal increase caused by additional frequency parameters.

For point cloud Transformers, the situation is more nuanced due to the architecture-specific attention design. Standard Point Transformer variants employ *vector attention*, where attention weights are computed per channel and combined with relative positional encodings. When nD-RoPE is applied under this setting, the computational cost remains comparable to the baseline, as the core attention structure is preserved. Notably, nD-RoPE also enables a transition from vector attention to standard *dot-product attention*. This change significantly reduces computational complexity, as scalar attention weights are shared across channels, leading to a substantial reduction in FLOPs. However, as discussed in the main paper, this efficiency gain comes at the cost of a slight performance drop in segmentation accuracy.

*Table 7.* Ablation study of the RoPE frequency base $\theta$ in nD-RoPE with vector attention under varying input point densities on ModelNet40. Models are trained with 2048 input points, and instance-average mIoU (%) is reported.

| Input Points | Base = 2 | Base = 100 | Base = $10^4$ | Base = $10^6$ |
|---|---|---|---|---|
| 256 | 53.59 | 55.04 | 45.94 | 46.16 |
| 768 | 78.05 | 78.80 | 74.29 | 78.74 |
| 1024 | 82.20 | 82.66 | 80.55 | 81.62 |
| 1536 | 85.43 | 85.61 | 85.17 | 82.67 |
| *2048* | 85.80 | 85.58 | 83.66 | 83.57 |
| 3072 | 84.78 | 84.98 | 84.70 | 84.00 |
| 4096 | 82.47 | 82.88 | 82.77 | 82.09 |

*Table 8.* Comprehensive Computational Cost Analysis. Comparisons across Image, Point Cloud, and Video tasks. For Point Transformers, we explicitly compare Vector Attention vs. Dot-Product Attention variants. Note that applying nD-RoPE with Vector Attention increases cost, but enabling Dot-Product Attention (Ours) significantly reduces FLOPs.

| Task | Backbone | PE Method | Attention Type | FLOPs (G) | Params (M) |
|---|---|---|---|---|---|
| Image | DeiT-S | Original (Learnable PE) | Dot-Product | 4.61 | 22.06 |
| | | nD-RoPE | Dot-Product | 4.89 | 23.36 |
| Video | TimeSformer | Original (Learnable PE) | Dot-Product | 196.05 | ~121 |
| | | nD-RoPE | Dot-Product | 196.05 | ~121 |
| Point Cloud | Point Trans. v1 | Original (Learnable Rel. PE) | Vector | 36.72 | 19.40 |
| | | nD-RoPE | Vector | 37.49 | 19.40 |
| | | nD-RoPE | Dot-Product | 13.89 | 14.14 |
| | Point Trans. v2 | Original (Learnable Rel. PE) | Vector | 50.04 | 11.32 |
| | | nD-RoPE | Vector | 51.06 | 11.33 |
| | | nD-RoPE | Dot-Product | 16.19 | 11.09 |

Overall, these results demonstrate that nD-RoPE itself introduces negligible computational overhead, and that the observed FLOPs variations in point cloud models primarily stem from the choice of attention formulation rather than the positional embedding mechanism.

## E. Optimal Scale Ratio Under an Economy Principle

We seek the optimal ratio between adjacent spatial scales that minimizes the total representation cost in $n$-dimensional Euclidean space. This yields a direct constraint on the geometric base used by $n$-dimensional rotary positional embeddings.

**Setup.** Consider $m$ discrete scales with periods $\{\lambda_i\}_{i=1}^m$ (larger $i$ is coarser) and activation diameters $\{l_i\}_{i=1}^m$ (abstractly, resolution limits). We normalize the global unit by setting the coarsest reference to $\lambda_0 = 1$, so every period is measured relatively. Let the adjacent ratio be $r_i = \lambda_{i+1}/\lambda_i > 1$ and $r_1 = \lambda_1/l_1$. To avoid aliasing across scales, a finer scale must not fold within the coarser activation diameter, which enforces

$$l_i \leq \lambda_{i-1} \quad \Longrightarrow \quad \left(\frac{\lambda_i}{l_i}\right)^n \geq r_i^n. \tag{30}$$

Here $n$ denotes the dimensionality of the underlying space (e.g., $n=2$ for images, $n=3$ for 3D data). The exponent reflects that coverage and overlap are measured volumetrically across all $n$ spatial dimensions—each axis contributes a factor of $\lambda_i/l_i$. The geometric feasibility constraint $l_i \leq \lambda_{i-1}$ ensures that finer modules do not alias within the activation span of coarser ones. Fig. 8 visualizes this relationship, where violating the condition leads to spatial ambiguity across scales.

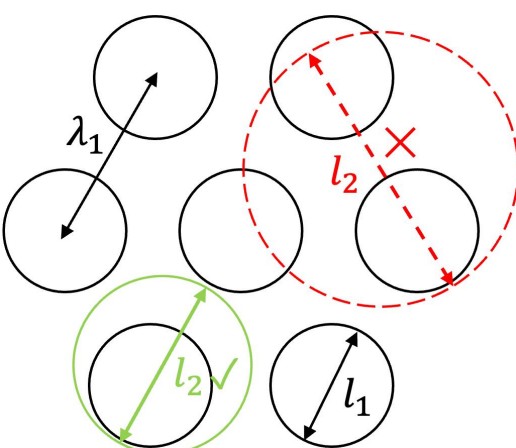

*Figure 8.* Illustration of the geometric feasibility condition between adjacent scales. Each circle represents the activation extent at a certain scale. If the finer-scale diameter $l_2$ (red) exceeds the coarser period $\lambda_1$, spatial ambiguity occurs (red cross). When $l_2 \leq \lambda_1$, as in the green case, the hierarchy remains uniquely decodable.

**Cost and target resolution.** To cover $\mathbb{R}^n$ without overlaps at scale $i$, roughly $(\lambda_i/l_i)^n$ distinct phase-shifted elements are required. Assuming at least $d$ such components per point, the total cost can be approximated as

$$N = \sum_{i=1}^{m} d \left( \frac{\lambda_i}{l_i} \right)^n. \tag{31}$$

A fixed global resolution $R$ is defined by how many times the smallest resolvable unit fits within the largest period—linking the coarsest scale to the finest. When all scales follow a constant geometric ratio $r_i \equiv r$ (a geometric ladder), this relationship simplifies to

$$R = (\frac{\lambda_m}{l_1})^n = (r^n)^m. \tag{32}$$

**Lower bound and optimal ratio.** By (30), each term in (31) is bounded below by $r_i^n$, so with $r_i \equiv r$ and (32) we obtain

$$N \geq d\, m\, r^n = d\, \rho\, \log_\rho R, \qquad \rho \triangleq r^n. \tag{33}$$

For fixed $R$, the function $\rho \log_\rho R$ over $\rho > 1$ is uniquely minimized at $\rho = e$. Therefore,

$$r^\star = e^{1/n}. \tag{34}$$

Intuitively, a geometric ladder with adjacent ratio $e^{1/n}$ achieves the target resolution with the least total coverage cost.

**Implication for geometric-frequency embeddings.** Consider a $n$-dimensional rotary position embedding with geometric frequencies

$$\omega_k = \text{base}^{-\frac{2k}{d}}, \qquad k = 0, 1, \ldots, \frac{d}{2} - 1, \tag{35}$$

so adjacent frequency slots form a constant ratio. Suppose one *scale* groups $M$ such frequency pairs before moving to the next scale, then the induced inter-scale ratio is:

$$\mathcal{R} = \text{base}^{\frac{2M}{d}}. \tag{36}$$

To be compatible with the optimal geometric ladder (34), we require

$$\mathcal{R} \leq e^{1/n} \quad \Longrightarrow \quad \text{base} \leq e^{\frac{d}{2Mn}}. \tag{37}$$

**Normalization and practical scaling.** The bound (37) is derived under the unit normalization $\lambda_0 = 1$, i.e., in a dimensionless domain. In practice, if the smallest distinguishable physical change is $\Delta_0$ (e.g., $\Delta_0 = 1$ pixel for images, $\Delta_0 = 0.01$ for point clouds in meters, or $\Delta_0 = 1/16000$ for audio), the same economy principle applies after normalizing lengths by $\Delta_0$. Denoting by base$^\star$ the optimal base in the normalized domain, the effective base in physical units becomes

$$\text{base}_{\text{eff}} = (\text{base}^\star)^{\lambda_0/\Delta_0} = (\text{base}^\star)^{1/\Delta_0} \quad (\lambda_0 = 1), \tag{38}$$

which preserves the same geometric spacing of frequencies in absolute units.

**Summary.** An economy-driven geometric ladder admits the unique optimal adjacent ratio $r^\star = e^{1/n}$. For a geometric frequency grid with embedding dimension $d$ and $M$ frequency pairs per scale, the positional-embedding base should satisfy (37). When moving from a normalized domain to physical units with minimal interval $\Delta_0$, adjust the base following (38).

