# OpenReview forum: "nD-RoPE: A Generalized RoPE for n-Dimensional Position Embedding"
_ICML.cc/2026/Conference — ICML 2026 regular_

### Official Review · Reviewer_fiBd · 2026-02-17

**Soundness:** 2
**Presentation:** 2
**Significance:** 1
**Originality:** 1
**Overall Recommendation:** 2
**Confidence:** 4

**Summary:**

nD-RoPE, a generalized $n$-dimensional RoPE, is proposed in this manuscript. Instead of leveraging axis decomposition or empirical frequency mixing, nD-RoPE leverages Fourier transform and Fourier expansion. Meanwhile, authors provide some theoretical analysis and wave vector selection strategy for their proposed nD-RoPE. Experiments on several benchmarks indicate the superior performance of nD-RoPE over its several counterparts.

**Compliance With Llm Reviewing Policy:**

Affirmed.

**Final Justification:**

I carefully read authors' rebuttal, their provided anonymous material via the anonymous link, and reevaluate their manuscript. However, I have to say that my concerns have not been addressed.

1, Based on authors' current rebuttal response and a careful reevaluation of FoPE and authors' nD-RoPE, my current conclusion is that the main mathematical insight of FoPE and nD-RoPE are still similar. If we look back to authors' manuscript, we can find that just as authors claimed in the first introduction paragraph of Section 4, the key to so-called "nD" applicability is directly derived by a key term $e^{j\omega^{T} x}$. With this term,  based on Fourier analysis and other (fairly basical) theoretical derivations, authors' nD-RoPE was derived. However, a similar key term has been proposed in FoPE. In the FoPE paper, the key term for the Fourier analysis is the similar $e^{j\omega^{T} n}$. So, from this perspective, the FoPE and RoPE are technically similar, let alone the latter Fourier analysis of both nD-RoPE and FoPE.

2, More seriously, I am very surprised by the following admitted sentence in the authors' rebuttal response.

> FoPE is indeed an excellent paper that we have closely followed during the development of nD-RoPE.

If authors closely followed the FoPE when authors developed nD-RoPE, **then why did not authors cite FoPE in their initial manuscript?** Authors' claimed different targets do not provide an appropriate reason for such critical omission given the technical similarity of FoPE and nD-RoPE. The cited RoPE-Axial variants do not target at so-called $n$ dimension as well, but authors still cite and compare their nD-RoPE with them. Initially, I think maybe authors do not know the existence of nD-RoPE, but I now realize I was too positive. **It means that the omission of FoPE was intended, and such intended omission could mislead reviewers to give a high score without knowing the existence of existing FoPE.** Considering the similar form of key terms of FoPE and nD-RoPE and many similar mathematical techniques, such intended omission is hard to be accepted.

3, I asked authors the following question in my initial review.

>Could you conduct additional experiments to show the benefits of nD-RoPE over FoPE? I suggest authors conduct the same experiments as FoPE without change any hyper-parameter.

However, authors do not provide such comparison. Authors implemented FoPE by themselves, and even compared with the authors' implemented FoPE, **RoPE alone cannot outperform FoPE in many important data metrics**. This phenomenon makes me doubt that although the FoPE did not primarily target at the so-called $n$ dimension, in each axis of FoPE, multiple frequencies are considered in each axis via the term $e^{j\omega n}$, and this may be enough for many tasks.

4, Currently, I doubt the significance of nD-RoPE. Authors' experiments indicate that using nD-RoPE alone is not the most favorable optiion, and they admitted it in their rebuttal response. From authors' current experiments, the significant improvement of nD-RoPE can only be shown via the combination of YaRN. However, it should be pointed out that YaRN is not the mandatory module for every RoPE task. This makes me doubt the real significance of nD-RoPE.

5, I hoped authors provide runtime metrics beyond the reported FLOPs, as different types of floating number operations may have different time costs. Authors misunderstood my point.

6, As I commented in my initial review,
>Authors do not conduct experiments or rigor theoretical analysis to compare their nD-RoPE with the following RoPE algorithm which is also applicable to $n$-dimension.
>
>Rethinking RoPE: A Mathematical Blueprint for N-dimensional Rotary Positional Embedding. URL: https://arxiv.org/abs/2504.06308.

However, authors comparision is still too empirical without supporting by any theoretical analysis and data.

Therefore, in the current stage, I have to lower my recommended scores. I choose (b) in this acknowledgement because I want to offer authors another opportunity to address my concerns. If authors' further response in the discussion period resolves my concerns, I am still willing to raise my scores.

**Key Questions For Authors:**

I have the following important questions. I will carefully consider authors responses in the rebuttal phrase. I am very willing to change my recommended scores if it is necessary.

1, What is the most fundamental difference of the ideas of nD-RoPE and those of FoPE? Could you conduct additional experiments to show the benefits of nD-RoPE over FoPE? I suggest authors conduct the same experiments as FoPE without change any hyper-parameter. You can see FoPE and experiments of FoPE by the following URL.

Fourier Position Embedding: Enhancing Attention's Periodic Extension for Length Generalization. ICML 2025. URL: https://openreview.net/forum?id=ZfDNDkg7Dh

2, Could you conduct experiments comparing the efficiency of different embedding methods?

3, Could you conduct theoretical analysis and experiments to compare nD-RoPE and the following RoPE algorithm?

Rethinking RoPE: A Mathematical Blueprint for N-dimensional Rotary Positional Embedding. URL: https://arxiv.org/abs/2504.06308.

4, Could you provide more detailed and rigorous analysis of the reasons why directly applying nD-RoPE is not that favorable? In Table 1 and Table 2, directly applying nD-RoPE without YaRN does not seem to be favorable.

**Limitations:**

Yes

**Strengths And Weaknesses:**

**Strengths:**

(1) The idea of introducing Fourier transforms and Fourier expansions to obtain the $n$-dimensional RoPE is sound.

(2) From an overall perspective, the writing and presentation of this manuscript is good. This manuscript is very easy to follow.

(3) Experiments over various benchmarks indicate the promising applications of nD-RoPE.

**Weaknesses:**

(1) The introduction of Fourier transform and Fourier expansion to RoPE is the main idea of nD-RoPE. However, Fourier position embedding (FoPE) has been proposed in ICML 2025:

Fourier Position Embedding: Enhancing Attention's Periodic Extension for Length Generalization. ICML 2025. URL: https://openreview.net/forum?id=ZfDNDkg7Dh

Meanwhile, in the current manuscript, FoPE is not discussed or compared. I notice that there may be some differences between the ideas of nD-RoPE and FoPE, but detailed discussion and comparison is necessary. Meanwhile, I think the rooted idea of nD-RoPE is somewhat very similar with FoPE: Both of nD-RoPE and FoPE introduce a new term $e^{i\omega x}$ and conduct Fourier expansion for this new term. This term is the foundation of benefits of nD-RoPE.

(2) Efficiency is an important factor to compare different embedding methods. Although authors admit nD-RoPE needs more computation, the experiments of current manuscript do not compare the efficiency of different embedding methods.

(3) Authors do not conduct experiments or rigor theoretical analysis to compare their nD-RoPE with the following RoPE algorithm which is also applicable to $n$-dimension.

Rethinking RoPE: A Mathematical Blueprint for N-dimensional Rotary Positional Embedding. URL: https://arxiv.org/abs/2504.06308.

(4) From Table 1 and Table 2, it seems that directly applying nD-RoPE is not stronger than some of its counterparts.




**Minor Comments:**

The legends of Fig. 2 are too small. I suggest authors enlarge the legends of Fig. 2.

---

> ### Author Rebuttal · Authors · 2026-03-30
>
> Thank you for the thoughtful questions. Please see the anonymous supplementary [FoPE and efficiency results [here]](https://anonymous.4open.science/r/nD-RoPE-26B9/reb.pdf).
>
> **[W1+Q1] Comparison with FoPE**
>
> Thanks for the suggestion. FoPE is indeed an excellent paper that we have closely followed during the development of nD-RoPE. The most fundamental difference is that nD-RoPE is a positional embedding design for joint n-dimensional space, whereas FoPE mainly targets length extrapolation in standard RoPE by modifying the training frequency spectrum. We therefore did not include FoPE in the original submission and instead used YaRN as the extrapolation baseline.
>
> That said, we agree that this comparison is informative, and we therefore conducted additional experiments on image and video datasets under the same protocol. We evaluate FoPE and FoPE+YaRN, implemented on top of RoPE-Axial using FoPE’s Fourier-series mechanism. Representative results are shown below, while the corresponding nD-RoPE and other variant results are reported in Tables 1–2 of the main paper.
>
> |Res|Img FoPE|Img FoPE+YaRN|Vid FoPE|Vid FoPE+YaRN|
> |-|-|-|-|-|
> |224|79.91|79.91|75.10|75.10|
> |448|74.31|77.30|67.84|74.41|
> |768|56.08|65.43|49.30|65.72|
> |1024|32.62|53.46|33.40|55.87|
>
> Overall, FoPE is a strong extrapolation baseline and generally outperforms the other RoPE variants. On ImageNet-1K, it remains below nD-RoPE throughout the extrapolation range. On Kinetics-400, it becomes competitive at larger resolutions and slightly exceeds nD-RoPE at the far end without YaRN, while nD-RoPE+YaRN remains strongest overall. This suggests that FoPE is particularly strong for extrapolation, whereas nD-RoPE offers more consistent gains from its joint n-dimensional positional design.
>
> **[W2+Q2] Efficiency Comparison**
>
> Thank you for this suggestion. Appendix D.4 and Table 8 already compare the efficiency of nD-RoPE with the original architecture. The main takeaway is that nD-RoPE introduces only a small increase in FLOPs, with a comparable parameter count, since it mainly changes the sampling of $\omega$ and its combination with positional coordinates rather than adding substantial new learnable components. The same holds for the other RoPE variants: their FLOPs and parameter counts also increase only marginally. Complete comparison tables for all embedding methods are available at the anonymous link above.
>
> **[W3+Q3] Comparison with Recent N-dimensional RoPE Methods**
>
> Thanks for pointing this out. This is a highly relevant recent work, and we have already discussed it in the related-work section. Like LieRE, it studies higher-dimensional RoPE from a Lie-group and Lie-algebra perspective, characterizing valid N-dimensional RoPE constructions through relativity and reversibility, and introducing cross-dimensional interactions via a learnable orthogonal basis transform.
>
> The main difference from our work is that their framework emphasizes algebraic characterization and learnable basis transforms, whereas ours emphasizes a constructive geometric design based on coverage, isotropy, and minimality. For this reason, in Sec. 5 we group it with RoPE-Mixed-style variants, namely methods that introduce cross-dimensional interactions through learnable parameters, rather than with our fixed simplex-based construction. We will revise the paper to make this relationship and distinction more explicit in the related-work and comparison sections.
>
> **[W4+Q4] Why nD-RoPE Alone Is Weaker in Extrapolation**
>
> Thanks. We agree that directly applying nD-RoPE without YaRN is not the most favorable setting for extrapolation. The main reason is that nD-RoPE improves the multi-dimensional positional structure, but does not by itself correct the position-scaling mismatch between training and test ranges. YaRN addresses this issue by rescaling unseen positions back into a frequency range closer to training.
>
> Thus, nD-RoPE and YaRN play complementary roles: nD-RoPE improves the geometric organization of positional encoding, while YaRN improves long-range extrapolation. This is why, among RoPE-style variants without additional extrapolation mechanisms, nD-RoPE still shows the slowest performance degradation as the range increases, and with YaRN it remains among the strongest overall. This suggests that nD-RoPE provides a stronger positional structure, while YaRN addresses the extrapolation-scale mismatch.
>
> More generally, directly extrapolating RoPE-family embeddings can lead to spectral distortion due to in-range and out-of-range frequency mixing, which is also consistent with the FoPE analysis. In this sense, the limitation is not specific to nD-RoPE, but to RoPE-style extrapolation without an explicit scaling mechanism such as YaRN. We will revise Sec. 5.3 to make this distinction clearer.
>
> **[Minor Comments]**
>
> Thanks! We will enlarge the legend in Fig. 2.

---

> > ### Author Rebuttal · Reviewer_fiBd · 2026-04-02
> >
> > I carefully read authors' rebuttal, their provided anonymous material via the anonymous link, and reevaluate their manuscript. However, I have to say that my concerns have not been addressed.
> >
> > 1, Based on authors' current rebuttal response and a careful reevaluation of FoPE and authors' nD-RoPE, my current conclusion is that the main mathematical insight of FoPE and nD-RoPE are still similar. If we look back to authors' manuscript, we can find that just as authors claimed in the first introduction paragraph of Section 4, the key to so-called "nD" applicability is directly derived by a key term $e^{j\omega^{T} x}$. With this term,  based on Fourier analysis and other (fairly basical) theoretical derivations, authors' nD-RoPE was derived. However, a similar key term has been proposed in FoPE. In the FoPE paper, the key term for the Fourier analysis is the similar $e^{j\omega^{T} n}$. So, from this perspective, the FoPE and RoPE are technically similar, let alone the latter Fourier analysis of both nD-RoPE and FoPE.
> >
> > 2, More seriously, I am very surprised by the following admitted sentence in the authors' rebuttal response.
> >
> > > FoPE is indeed an excellent paper that we have closely followed during the development of nD-RoPE.
> >
> > If you closely followed the FoPE when you developed nD-RoPE, **then why did not you cite FoPE in your initial manuscript?** Authors' claimed different targets do not provide an appropriate reason for such critical omission given the technical similarity of FoPE and nD-RoPE. The cited RoPE-Axial variants do not target at so-called $n$ dimension as well, but authors still cite and compare their nD-RoPE with them. Initially, I think maybe authors do not know the existence of FoPE, but I now realize I was too positive. **It means that the omission of FoPE was intended, and such intended omission could mislead reviewers to give a high score without knowing the existence of existing FoPE.** Considering the similar form of key terms of FoPE and nD-RoPE and many similar mathematical techniques, such intended omission is hard to be accepted.
> >
> > 3, I asked authors the following question in my initial review.
> >
> > >Could you conduct additional experiments to show the benefits of nD-RoPE over FoPE? I suggest authors conduct the same experiments as FoPE without change any hyper-parameter.
> >
> > However, authors do not provide such comparison. Authors implemented FoPE by themselves, and even compared with the authors' implemented FoPE, **nD-RoPE alone cannot outperform FoPE in many important data metrics**. This phenomenon makes me doubt that although the FoPE did not primarily target at the so-called $n$ dimension, in each axis of FoPE, multiple frequencies are considered in each axis via the term $e^{j\omega n}$, and this may be enough for many tasks.
> >
> > 4, Currently, I doubt the significance of nD-RoPE. Authors' experiments indicate that using nD-RoPE alone is not the most favorable optiion, and they admitted it in their rebuttal response. From authors' current experiments, the significant improvement of nD-RoPE can only be shown via the combination of YaRN. However, it should be pointed out that YaRN is not the mandatory module for every RoPE task. This makes me doubt the real significance of nD-RoPE.
> >
> > 5, I hoped authors provide runtime metrics beyond the reported FLOPs, as different types of floating number operations may have different time costs. Authors misunderstood my point.
> >
> > 6, As I commented in my initial review,
> > >Authors do not conduct experiments or rigor theoretical analysis to compare their nD-RoPE with the following RoPE algorithm which is also applicable to $n$-dimension.
> > >
> > >Rethinking RoPE: A Mathematical Blueprint for N-dimensional Rotary Positional Embedding. URL: https://arxiv.org/abs/2504.06308.
> >
> > However, authors comparision is still too empirical without supporting by any theoretical analysis and data.
> >
> > Therefore, in the current stage, I have to lower my recommended scores. I choose (b) in this acknowledgement because I want to offer authors another opportunity to address my concerns. If authors' further response in the discussion period resolves my concerns, I am still willing to raise my scores.

---

> > > ### Author Response · Authors · 2026-04-04
> > >
> > > # Clarification
> > > Thank you for the follow-up. We apologize for the confusion caused by our previous wording. The omission of FoPE in the initial manuscript was not intentional. Rather, it reflected our scoping toward higher-dimensional positional encoding methods, whereas FoPE primarily targets 1D extrapolation and spectral correction. Accordingly, although FoPE is relevant, we did not initially consider it a direct baseline for our primary research question. We agree that FoPE should be cited for completeness, and we will revise the paper accordingly. To further clarify this positioning, we also evaluated FoPE under our setting and report the comparison in (3).
> > >
> > > ## [Q1–Q3]
> > > ### (1) Relationship between FoPE and nD-RoPE
> > > We agree that FoPE and nD-RoPE share some mathematical language (e.g., Fourier/complex formulations), and we appreciate the reviewer for pointing out this connection. However, the two works target different problems.
> > >
> > > FoPE focuses on 1D RoPE length extrapolation. It analyzes frequency mixing and spectral damage during propagation from a NUDFT perspective, and introduces a discrete-Fourier-transform-based correction to improve spectral stability and long-context generalization.
> > >
> > > In contrast, nD-RoPE addresses how to construct RoPE in joint n-dimensional space under translation invariance. Our derivation starts from the translation-invariance assumption and uses a random-Fourier-feature (RFF) perspective to obtain a unified product form for positional interactions in nD space.
> > >
> > > Thus, despite similarities in representation, the two works differ in their starting assumptions, problem formulation, and resulting contributions.
> > >
> > > ### (2) Reason for omission and revision plan
> > > Our original comparison scope focused on methods addressing similar objectives, namely positional encoding design in higher-dimensional settings. Since FoPE is primarily positioned as a solution to length extrapolation in 1D (and is evaluated against methods such as ALiBi), we did not initially treat it as a direct comparison baseline.
> > >
> > > That said, we agree that FoPE is relevant, especially from the perspective of frequency-domain analysis of RoPE. Its omission from the related work was a positioning oversight, not an intent to exclude prior work.
> > >
> > > We will revise the paper to: (i) cite and discuss FoPE in related work, (ii) clarify its relationship to nD-RoPE, and (iii) better contextualize our contribution relative to FoPE and other 1D extrapolation methods.
> > >
> > > ### (3) Experimental comparison
> > > As analyzed in Sec. 4.2, nD-RoPE reduces to standard RoPE in the 1D case. Thus, applying it directly to FoPE’s original 1D text setting would reduce to the standard RoPE baseline, rather than reflect behavior specific to nD positional design. Accordingly, our primary comparisons focus on higher-dimensional RoPE variants aligned with our setting, rather than 1D extrapolation methods.
> > >
> > > For completeness, we also evaluated FoPE in the higher-dimensional settings targeted by our paper. Following its released implementation, we applied its Fourier-series mechanism under the same protocol and reported the results. FoPE is strong for length extrapolation; however, nD-RoPE still performs better in most target-domain extrapolation settings.
> > > ## [Q4]
> > > Our experiments are split into two parts: one without extrapolation techniques to evaluate nD-RoPE itself, and one with techniques such as YaRN to test whether its gains persist when combined with standard RoPE extensions.
> > >
> > > Even without YaRN, Tables 1–2 and Fig. 5 already show clear gains for nD-RoPE, mainly in its slower degradation under scale shift. The YaRN results therefore indicate complementarity rather than dependence.
> > > ## [Q5]
> > > We also report below the 1-epoch training time (min:sec) of all PE methods on ImageNet-1K using a single A100.
> > >
> > > |Method|Time|
> > > |-|-|
> > > |FoPE|15:03|
> > > |Learnable PE|15:11|
> > > |Axial+APE|15:29|
> > > |Axial|15:30|
> > > |Mixed+APE|15:33|
> > > |Mixed|15:46|
> > > |nD-RoPE|16:56|
> > > ## [Q6]
> > > Theoretically, their framework starts from the general valid RoPE form:
> > > $$
> > > R_{x} = \exp\left(\sum_{i = 1}^{N} x_{i} B_{i}\right)
> > > $$
> > > where Lie-theoretic generators are constrained to satisfy the RoPE validity conditions. Under this formulation, the standard axis-wise RoPE appears as a valid special case.
> > >
> > > To introduce cross-dimensional interaction, they further use:
> > > $$
> > > R_{x} = \exp\left(\sum_{i = 1}^{N} x_{i} Q B_{i} Q^{T}\right)
> > > $$
> > > which is equivalent to applying a learnable orthogonal change of basis. This makes its practical mechanism very close to RoPE-Mixed-style learnable interaction, which is why we group it with RoPE-Mixed variants.
> > >
> > > Experimentally, the paper is largely theoretical and does not release open-source code. We therefore implemented the method ourselves and evaluated it on Kinetics-400, and the results are available at the anonymous link above. Without extrapolation, its performance falls between axial RoPE and mixed RoPE, consistent with the analysis above. We will clarify this in the revision.

---

### Official Review · Reviewer_2ft8 · 2026-03-10

**Soundness:** 2
**Presentation:** 2
**Significance:** 4
**Originality:** 3
**Overall Recommendation:** 5
**Confidence:** 3

**Summary:**

The paper proposes a general method for doing RoPE in $n$-dimensional spaces and verifies that it generalizes across geometric perturbations across a variety of datasets and tasks.

**Compliance With Llm Reviewing Policy:**

Affirmed.

**Final Justification:**

My primary concern regarding the theoretical framing has been largely addressed. I suggest that the authors include clear definition statements/lemmas/theorems which make their points in Section 4 clear. I have the impression from the rebuttal that they will do so.

**Key Questions For Authors:**

1. Would you be able to provide a formal lemma/theorem regarding what your method is able to do which other methods cannot?
2. What is YaRN and how does it impact the results? It seems necessary for interpreting the tables.
3. What are the results if you use different wave vector methods? The wave vector component feels like a "this is what seems to work best", but I don't see results in the main body of the paper which analyze this.

**Limitations:**

The paper presents a method which seems to work well, but due to the lack of formal lemmas and proofs, it is unclear to what extent the method outperforms other methods or, indeed, why one should expect it to. This limitation does not seem addressed in the paper to me. I would be very happy to raise my score if such a justification was provided.

**Strengths And Weaknesses:**

**Note that I am not particularly familiar with the RoPE literature (but am well-versed in kernel trick and Fourier transform ideas).**

## Strengths

The idea is elegant and natural. As I understand it, the main point is to effectively sample the $n$-dimensional space across frequencies and directions using Fourier methods. If this is indeed the first time that this idea has been introduced in this way then I am shocked and believe it to be a natural and necessary contribution. The results are convincing and the experiments are cleanly measuring for what the authors suggest their method should be better at. Really, I find it all quite compelling.

## Weaknesses

The presentation could really use some work, unfortunately. I believe that the ideas in this paper are good, but it is very difficult for me to verify this.

For one thing, I think that section 4 is all over the place and (to me) very difficult to follow. It feels like the authors are trying to backwards-justify a method which happens to work well by putting in a bunch of equations that are loosely related to one another. There is no proof showing the proposed method's effectiveness. Statements are made with zero motivation and terms are waved away with zero explanation. Here is an inexhaustive list of examples:
- Lines 196-199 (left column). Why is this obtained by "projecting a square-integrable function onto the orthonormal basis". Why $-x_1$ rather than $x_1$? What is the square integrable function and where does it come from? Why are we doing any of these steps?
- Lines 194-195 (right column). "We express $\gamma(q, x)$ via the inverse Fourier transform". Again... why? What is the purpose? What are we working towards? If there was a lemma statement we were trying to prove, it would be clear what we are trying to achieve with these otherwise arbitrary re-mappings of the terms.
- Line 219 (right column). This is the crux of the concerns. We have just spent 3/4 of a page working up to equation 13. Equation 13 now says that these two terms are "roughly equivalent" (as evidenced by the $\approx$ sign). But... how close is this equality? What is the bound on it? As it stands, without any bounds, we won't be able to make any claim about whether any of the above steps were useful.
- Line 220 (left column). "$W$ stacks the frequency-dependent weights $B^\top$." But we don't know what $B$ is? It just "exists" according to equation 9. So how am I supposed to use $W$ if it depends on a thing which we don't know?
- Lines 225-227 (left column). This is another crux of concerns. "In practice, $W$ can be absorbed". Why is this the case? There is no explanation for this at all? It just reads to me like the authors did a page worth of math to derive a thing which they then just sweep under the rug because they don't know how to work with it any further?

In general, this whole section was very confusing. I would recommend the authors make formal theorems about what the other methods are not able to do and what their method *is* able to do, with clear bounds establishing this discrepancy. Otherwise, I find that the math is distracting.

Separately, I also think a few other pieces of presentation could be improved. For example, the actual math behind RoPE was never introduced. I have no context for how RoPE works, so this made it hard to follow how the authors are changing it. Similarly, YaRN was never introduced and I have no idea what it is, but it seems useful towards understanding the results?

---

> ### Author Rebuttal · Authors · 2026-03-30
>
> Thank you for your feedback and careful review.
>
> **[Weakness] Clarifying the derivation in Sec. 4.1**
>
> We agree that Sec. 4.1 is dense and may read as a post-hoc justification. Our intention, however, is different: Sec. 4.1 provides a constructive derivation from translation-invariant relative-position attention to a finite-dimensional RoPE-style parameterization in $n$ dimensions. The goal is to use the spectral representation naturally implied by translation invariance to derive a practical finite-dimensional positional form, rather than to justify an already fixed design.
>
> On the specific points:
>
> - Lines 196–199. The projection step represents the feature at position $x_1$ in a basis translated to $x_1$, so that the resulting inner product depends on relative displacement. The shifted basis $\phi_i(\cdot-x_1)$ centers the basis at $x_1$; using $\phi_i(\cdot+x_1)$ would shift it in the opposite direction.
>
> - Lines 194–195. The inverse Fourier step is introduced because translation-invariant kernels admit a spectral representation. This allows the relative-position dependence to be rewritten in a form that separates content and position, making the positional dependence explicit in terms of frequencies and leading to the position-dependent feature map.
>
> - Line 219. Eq. (11) is the exact continuous spectral form, while Eq. (13) becomes approximate only because the continuous spectrum is replaced by finitely many sampled frequencies. Thus, “$\approx$” marks the explicit finite-feature approximation, rather than a vague heuristic step.
>
> - Lines 220, 225–227. $B(\omega)$ is the continuous coefficient function in the exact derivation. After discretization, its sampled values become a finite matrix $W$, which can be absorbed into the standard learned query projection $W_Q$. This is why the continuous derivation and the practical implementation are equivalent rather than inconsistent.
>
> We will revise Sec. 4.1 to clarify the roadmap, the role of the auxiliary function-space representation, and the approximation from Eq. (11) to Eq. (13).
>
> **[Q1] On Formal Theoretical Guarantees**
>
> Thank you for this suggestion. We agree that a clearer formalization would help clarify the theoretical contribution. Our goal in Sec. 4 is not to establish a strict superiority theorem, but to provide a principled derivation. The key formalization is best understood in terms of two structural statements: (i) a representation result showing that, under the translation-invariance assumption, the positional embedding admits a Fourier product form $\gamma(q,x)=q^T\phi(x)$, with Eq. (13) as its finite-frequency approximation; and (ii) a geometric result showing that the regular-simplex wave-vector construction yields a minimal full-coverage design with isotropic directional structure, in contrast to axis-wise constructions.
>
> This is made explicit in Sec. 4.3, where the final nD-RoPE construction (Eq. 16) provides a closed-form finite-dimensional embedding. In this sense, Sec. 4.1 explains why the Fourier form arises, Sec. 4.2 determines how the wave vectors are chosen, and Sec. 4.3 specifies the resulting embedding. We will revise Sec. 4 to make these structural roles more explicit.
>
> **[Q2] RoPE and YaRN Clarification**
>
> We agree that a brief introduction to RoPE and YaRN would improve readability, especially for readers less familiar with this line of work. Due to space constraints, we focused on the motivational analysis and did not introduce these preliminaries explicitly. We will add a short clarification in the revision.
>
> **[Q3] Wave-Vector Design: Heuristic or Principled?**
>
> We agree that the current presentation may give the impression that the wave-vector design is heuristic and that comparisons with alternative constructions are not sufficiently emphasized in the main text.
>
> The choice of wave vectors is not based on empirical preference alone, but follows from three geometric constraints: (i) full coverage, so that the positional representation spans the full spatial domain; (ii) isotropy of the induced real-space representation, so that no spatial direction is privileged a priori; and (iii) minimality, so that representational budget is not spent on redundant directions and can instead be allocated to multiple scales. Under these requirements, the simplex arises as a principled solution rather than an ad-hoc design.
>
> In the current paper, Appendix E derives the relationship between frequency bases and scales from the economy principle, while Appendix D.3 analyzes how different frequency bases affect the resulting wave-vector construction. In addition, in our rebuttal to Reviewer `AdnE [W3]`, we further examine the sensitivity of the regular-simplex design under small perturbations of $\omega$. Together, these analyses support the geometric rationale behind the regular-simplex construction, rather than a single hand-picked choice.
>
> We will revise Sec. 4.2 to make this reasoning and the relevant comparisons more explicit.

---

> > ### Author Rebuttal · Reviewer_2ft8 · 2026-04-01
> >
> > I appreciate the points and clarifications regarding the text in Section 4. I also appreciate the clarification on the approximation step and the absorption of $B(\omega)$; these points are now clearer to me. As I said in my review, I find the ideas compelling.
> >
> > However, I'd really like to push a bit on the formalization component via theorems/lemmas/etc. Given that the paper frames its contribution as a theoretical foundation, I believe formal statements are warranted. Otherwise, the work risks being ambiguous in what the specific claim is and what the evidence for it is.
> >
> > The abstract suggests that the authors' method "provides uniform directional coverage with maximal symmetry". They do this by "deriving a spectral condition for isotropy that requires treating positions and frequencies as coupled n-dimensional vectors". These are reasonable points. What I am requesting is that the authors provide formal theorem/lemma statements (with step-by-step proofs) making this explicit. For example, such a statement could be something like: "For any axis-wise construction with $M$ frequencies in $n$ dimensions, there exist directions along which the representation degenerates, whereas the simplex construction with n+1 vectors achieves uniform coverage." Here, I am assuming formalized definitions of "degenerating" and "uniform coverage" which would need to be provided accordingly. But this is just a suggestion rather than a specific request -- I am not sure what the best theoretical statement would be which supports the claims the authors are making.
> >
> > To be clear, I am not suggesting that the authors should make these theorems to verify "superiority" of their method as was suggested in the rebuttal. Instead, I am asking for a formal proof that other methods fail at a specific task whereas the authors' method succeeds at it. As it currently stands, it is not clear to me what, precisely, other methods fail at and in what way, specifically, the authors' method makes an improvement.
> >
> > If you are able to provide such statements without relying on unreasonable or overly simplistic assumptions, then I think it would strengthen the paper considerably. Otherwise, I am inclined to maintain my score.
> >
> > Thank you again!

---

> > > ### Author Response · Authors · 2026-04-02
> > >
> > > ### **Clarifying the Scope of the Theoretical Contribution**
> > >
> > > Thank you for this helpful comment.
> > >
> > > We agree that we did not make the distinct roles of Secs. 4.1 and 4.2 sufficiently clear. In the introduction, the phrase “theoretical foundation” in our second contribution was intended to refer to the derivation in Sec. 4.1. Starting from translation-invariant relative-position attention, Sec. 4.1 derives a unified $n$-dimensional Fourier product-form representation, which naturally leads to a finite-dimensional RoPE-style approximation. Sec. 4.2, by contrast, instantiates this formulation through a geometric wave-vector design rather than serving as the main derivational contribution.
> > >
> > > In the revision, we will make this distinction explicit in two ways: first, by describing the contribution of Sec. 4.1 more precisely as a principled theoretical formulation for unified $n$-dimensional positional encoding, rather than using the broader phrase “theoretical foundation”; and second, by clarifying that Sec. 4.2 is the geometric instantiation of this formulation.
> > >
> > > ---
> > >
> > > ### **Formal Statements for Sec. 4.2**
> > >
> > > We also agree that Sec. 4.2 should be made more formal.
> > >
> > > The main point we intend to make in this section is structural: a useful wave-vector design should induce a real-space positional pattern that satisfies two requirements, namely full-dimensional non-degenerate coverage and directional balance. The first rules out constructions that leave some spatial directions unresolved, while the second rules out constructions that, although full-dimensional, still privilege some directions over others. In the revision, we will make this structure explicit through a lemma capturing the failure mode of rank-deficient coverage, and a theorem capturing the corresponding directional-balance property of the centered regular-simplex design.
> > >
> > > **Lemma 1 (Rank deficiency implies directional degeneracy).**
> > >
> > > Let $\Omega \in \mathbb{R}^{M \times n}$ be the wave-vector matrix in Eq. (15). If $\mathrm{rank}(\Omega) < n$, then there exists a nonzero vector $v \in \mathbb{R}^{n}$ such that $\Omega v = 0$. Hence $\Omega(x + tv) = \Omega x$ for all $t \in \mathbb{R}$, so displacements along $v$ are not distinguished by the induced phase system. Therefore, to distinguish displacements in all spatial directions, it is necessary that $\mathrm{rank}(\Omega) = n$.
> > >
> > > **Theorem 1 (Regular simplex yields direction-independent second-order response).**
> > >
> > > Let $\Omega = \{\omega_i\}_{i=1}^{n+1} \subset \mathbb{R}^{n}$ be the centered regular-simplex wave-vector set, as constructed in Appendix A. In particular, it satisfies the standard simplex identities:
> > >
> > > $$
> > > \sum_{i=1}^{n+1} \omega_i = 0
> > > $$
> > >
> > > $$
> > > \|\omega_i\| = r
> > > $$
> > >
> > > $$
> > > \langle \omega_i, \omega_j \rangle = -\frac{r^2}{n}, \qquad \text{for } i \ne j
> > > $$
> > >
> > > Define
> > > $$
> > > S = \sum_{i=1}^{n+1} \omega_i \omega_i^\top.
> > > $$
> > >
> > > Using the simplex identities, one obtains
> > > $$
> > > S\omega_j
> > > = \left(\sum_{i=1}^{n+1} \omega_i \omega_i^\top\right)\omega_j
> > > = \sum_{i=1}^{n+1} \omega_i (\omega_i^\top \omega_j)
> > > = r^2 \omega_j - \frac{r^2}{n} \sum_{i \ne j} \omega_i
> > > = \frac{n+1}{n} r^2 \omega_j,
> > > \qquad \forall j \in \{1, \ldots, n+1\}.
> > > $$
> > >
> > > Here we used $\sum_{i \ne j} \omega_i = -\omega_j$. Since the centered simplex vectors span $\mathbb{R}^{n}$, it follows that
> > > $$
> > > S = \frac{n+1}{n} r^2 I_n.
> > > $$
> > >
> > > Therefore, for every unit vector $u \in \mathbb{R}^{n}$ with $\|u\| = 1$,
> > > $$
> > > \sum_{i=1}^{n+1} (\omega_i^\top u)^2
> > > = u^\top \left(\sum_{i=1}^{n+1} \omega_i \omega_i^\top\right) u
> > > = u^\top \left(\frac{n+1}{n} r^2 I_n\right) u
> > > = \frac{n+1}{n} r^2.
> > > $$
> > >
> > > Hence the regular-simplex wave-vector set assigns the same total second-order directional energy to every spatial direction, i.e., $\sum_{i=1}^{n+1} (\omega_i^\top u)^2$ is constant for every unit direction $u$. Equivalently, at the second-order level, the induced inner-product response to relative displacement is equal across spatial directions. This is the operational sense in which the construction is isotropic.
> > >
> > > ---
> > >
> > > ### **Summary**
> > >
> > > Taken together, these two statements make the intended logic of Sec. 4.2 explicit: Lemma 1 shows how an alternative wave-vector design can fail by leaving some real-space directions unresolved, while Theorem 1 shows how the centered regular-simplex design succeeds by distributing the same total second-order directional energy across all directions.
> > >
> > > We appreciate this valuable suggestion, and agree that making these structural roles explicit substantially strengthens the formal grounding of Sec. 4.2.

---

### Official Review · Reviewer_ADnE · 2026-03-11

**Soundness:** 4
**Presentation:** 3
**Significance:** 4
**Originality:** 3
**Overall Recommendation:** 4
**Confidence:** 3

**Summary:**

This paper proposes a unified theoretical framework for extending Rotary Position Embeddings (RoPE) to arbitrary n-dimensional Euclidean spaces. The authors identify a directional bias in existing multi-dimensional RoPE implementations, which treat spatial axes independently and fragment coherent geometric displacements, or have collapsed frequencies that limit the isotropic feature learning. The proposed method uniformly treats all spatial dimensions, building on top of a Hilbert-space framework, and embeds the n-dimensional offsets in a complete and isotropic manner by selecting wave vectors on a regular simplex. The experimental results show the proposed n-dim RoPE achieves higher performance than existing alternatives in image, video, and point cloud learning tasks, and also has better robustness against perturbations in rotation, resolution, and density.

**Compliance With Llm Reviewing Policy:**

Affirmed.

**Final Justification:**

To a large extent, the authors clarified Section 4.1, which is the derivation of the positional embedding construction. If I understand correctly, this construction is not a proof that all positional embeddings satisfying translational invariance have to follow the proposed form. But this point is unclear in the manuscript, and I couldn't verify if it will be addressed properly. Given that this is the core foundation of this paper, I couldn't raise the score, and will keep the current weak accept rating.

**Key Questions For Authors:**

1. Can you explain in more detail to fill the gaps in the derivations of Section 4.1? What's the big picture of this derivation? Is this theoretical derivation all necessary for introducing the $\omega^T x$ format of nD-RoPE? Is there a simpler way to introduce the foundation for the $\omega^T x$ format?

2. In the experiment, the rotation robustness is checked with 0-90 degree rotations. Will a rotation of 120 degrees, which aligns with the simplex wave vectors, have a better performance than angles in between?

3. What are the limitations of this work?

**Limitations:**

The authors didn't talk about the limitations.

**Strengths And Weaknesses:**

Strength:

1. Well-motivated problem and clear empirical contribution. The paper addresses a fundamental issue in modern machine learning. Suboptimal position embeddings inject wrong biases and hurt learning. The NUFT reconstruction visualizations (Figure 2) effectively illustrate the isotropy advantage. The simplex-based solution is elegant and intuitive.

2. Strong and comprehensive experiments. The evaluation spans diverse modalities (2D images, video, 3D point clouds) and stress-tests extrapolation beyond the training scale. nD-RoPE shows consistent advantages.

3. The simplex wave vector construction is principled and novel. The coverage and symmetry criteria in Section 4.2 are clearly stated, and the regular simplex is a well-justified solution that is also simple to implement.

4. Negligible computational overhead. Table 8 demonstrates that nD-RoPE introduces very small extra FLOPs or parameters for image and video transformers, making adoption straightforward.

Weakness:

1. The theoretical derivation in Section 4.1 is hard to follow and somewhat loose. A major part of this paper is the principled derivation of the $e^{j\omega^\top x}$ positional encoding form. However, the presentation is not very clear. The authors appear to address a general area of translation-invariant kernel design, but the construction $f_i(q, x_1) = \langle \gamma(q, \cdot), \phi_i(\cdot - x_1)\rangle$ is introduced without clear motivation. How do we know a function can be written in this way? Is it an assumption or derivation from what's given? Are f(q, x_1) and f(k, x_1) elements of the Hilbert space, or really just $\gamma(q, \cdot)$? What's the big picture of this construction? These all make this section inaccessible to readers without a strong background in functional analysis. Besides, the derivation from the other direction: showing that the $\omega^\top x$ form leads to translation invariance, is much easier. Including it or citing it from another paper will make it more intuitive for readers to understand the value of this design.

2. The $\omega^\top x$ formulation is not new, but this is underacknowledged. It already appeared in RoPE-Mixed, STRING, and others. The paper acknowledges these in the related work but does not clearly delineate what is new versus existing when introducing the $\omega^\top x$ formulation in Section 4. A reader could easily mistake the formulation itself as the paper's contribution rather than the wave vector selection strategy.

3. Section 4.1 (existing) occupies significantly more space than Section 4.2 (the novel contribution). This imbalance obscures what is new. The simplex construction could be motivated more gracefully and experimented with in more detail. For example, a more thorough discussion of why simplex is optimal among all possible configurations, how sensitive performance is to perturbations away from the simplex, and how the construction scales in very high dimensions would all strengthen the paper.

---

> ### Author Rebuttal · Authors · 2026-03-30
>
> Thank you for your feedback and careful review. Our responses are below. Additional complete results for [W3] and [Q2] are available [here](https://anonymous.4open.science/r/nD-RoPE-26B9/reb.pdf).
>
> **[W1+Q1] Clarification of the Derivation in Sec. 4.1**
>
> Thanks for this thoughtful comment. Sec. 4.1 mainly asks: if we assume only translation invariance, what positional encoding form follows? We therefore use a constructive route: translation invariance $\rightarrow$ functional lifting $\rightarrow$ Fourier spectral form $\rightarrow$ finite-frequency approximation. We agree this motivation was not sufficiently explicit and will clarify it at the start of Sec. 4.1. Starting from $e^{j\omega^T x}$ is more intuitive, but it is closer to a verification of a presumed form than a derivation from the translation-invariance assumption itself.
>
> Regarding the construction $f_{i}\left( q,x_{1} \right) = < \gamma(q, \cdot ),\phi_{i}( \cdot - x_{1}) >$, this is an intermediate functional representation, not an additional modeling assumption. Here, $\gamma(q, \cdot )$ is the lifted function-space representation of the content vector q, while $f_{i}(q,x_{1})$ is the scalar coefficient obtained by projecting onto the translated basis function $\phi_{i}( \cdot - x_{1})$. This step keeps the derivation in functional form, so translation invariance can be expressed spectrally before the finite-frequency approximation in Eq. (13).
>
> Sec. 4.1 is thus a bridge from translation invariance to the practical RoPE-style form. If our aim were only to motivate $\omega^T x$, a simpler reverse-direction introduction would suffice. Instead, we seek to derive it directly from the translation-invariance assumption. We will revise the section to clarify this roadmap, the role of the lifted representation, and the approximation step.
>
> **[W2] Novelty of the $\omega^T x$ formulation**
>
> Thanks for the comment. We agree that the cross-dimensional $\omega^T x$ form itself is not new, as stated in Sec.2. Our contribution in Sec. 4 is twofold. First, prior work typically starts from per-dimension constructions and introduces cross-dimensional interactions heuristically, whereas Sec. 4.1 derives an n-D RoPE-style form directly from the translation invariance assumption. Second, prior implementations often use learnable parameters for cross-dimensional interactions, whereas Sec. 4.2 provides a principled wave-vector design based on symmetry and isotropy. These two parts are complementary and together constitute the main contribution of Sec. 4.
>
> **[W3] Simplex Construction Motivation**
>
> We agree that, although Secs. 4.1 and 4.2 are complementary, the motivation for the simplex design should be presented more clearly. As clarified in our response to Reviewer `FWgu [Q3]+[Q4]`, we do not claim that the regular simplex is globally optimal over all possible configurations; rather, it is the natural choice under a minimal full-dimensional design. Under three constraints—full coverage, isotropy, and minimality—the regular simplex is the unique equiangular construction in $n$ dimensions and provides a unified design across dimensions.
>
> We also agree that perturbation sensitivity and high-dimensional scaling deserve clearer discussion. Additional checks with random perturbations of different magnitudes on $\omega$ suggest that the simplex design is stable under small perturbations, while extrapolation degrades gradually as perturbations increase, especially at larger resolutions. For high-dimensional scaling, we will clarify the limitation that since each scale uses n+1 wave vectors, increasing the ambient dimension reduces the number of scales representable under a fixed embedding budget.
>
> **[Q2] Simplex Alignment Effect**
>
> Thanks for this question. Based on our results, we do not observe a special performance peak at 120°. In the 120° to 180° range, the accuracies at 120°, 150°, and 180° are 72.59, 71.69, and 75.59. The main recovery occurs around 180°, not 120°. The trend is approximately symmetric from 180° to 360°.
>
> We therefore do not attribute the robustness to a simplex-aligned angle. The simplex wave vectors define directions in embedding frequency space, but they are not tied to the semantic orientation of the input image. A more plausible explanation is that the recovery near 180° reflects the visual symmetry and orientation statistics of natural images under rotation. Across these angles, nD-RoPE remains more robust than the baselines.
>
> **[Q3] Limitations**
>
> A main limitation of this work is that the derivation relies on the translation-invariance assumption in Euclidean space. As a result, the current framework is naturally suited to regular Euclidean domains such as images, videos, and point sets, but does not directly extend to irregular non-Euclidean domains such as graphs or curved manifolds. Extending the framework beyond Euclidean space is an important direction for future work.

---

> > ### Author Rebuttal · Reviewer_ADnE · 2026-04-04
> >
> > Thank you for the detailed rebuttal. I appreciate the authors' engagement with the review comments. Below are my concerns that haven't been fully addressed.
> >
> > W1/Q1 (Theoretical derivation in Section 4.1): The authors clarify the intended logical roadmap (translation invariance --> functional lifting --> Fourier spectral form --> finite-frequency approximation). However, the core gap I identified remains: the construction $f_i(q, x_1) = \langle \gamma(q, \cdot), \phi_i(\cdot - x_1)\rangle$ is described as "an intermediate functional representation, not an additional modeling assumption," but this does not explain why this particular form is the right intermediate step and it is still unclear to me if this construction introduces extra assumptions. As written, it seems to implicitly encode the cross-correlation structure it is supposed to derive.
> >
> > W2 ($\omega^\top x$ novelty): The authors clarify that their twofold contribution is (1) deriving the $\omega^\top x$ form from translation invariance, and (2) the simplex wave vector design. I appreciate this framing. However, as noted above, if contribution (1) remains logically incomplete after revision, then the primary contribution is effectively the simplex construction.
> >
> > Therefore, I maintain my current score.

---

> > > ### Author Response · Authors · 2026-04-04
> > >
> > > We thank the reviewer for raising this important point. Our intention is to clarify the relationship between content and position from the perspective of the attention mechanism itself. In attention, $q$ and $k$ first represent content, while positional information is introduced afterward. Motivated by this, rather than assuming a predefined fused form of content and position from the outset, we first seek a representation of content alone. Specifically, we lift each content vector $q$ to a function in a Hilbert space, denoted by $\gamma(q,\cdot)$, and then ask how position should act on this functional representation.
> > >
> > > Under this perspective, if the same content appears at different locations, what changes is not the content itself but its relative placement with respect to other tokens, and therefore its interaction pattern in attention. Under the translation-invariance assumption, a natural way to express this positional effect in function space is through translation:
> > >
> > > $$
> > > \gamma(q,\cdot)\mapsto \gamma(q,\cdot+x_1).
> > > $$
> > >
> > > Importantly, at this stage we do not impose a specific analytic form on $\gamma(q,x)$, and only specify that positional dependence enters the representation through translation.
> > >
> > > Accordingly, Eq. (4) is more clearly written as
> > >
> > > $$
> > > f_i(q,x_1)=\langle \gamma(q,\cdot+x_1),\phi_i(\cdot)\rangle,
> > > $$
> > >
> > > where $\{\phi_i\}$ is a common fixed basis. Under this form, $q$ selects the underlying function, $x_1$ introduces positional information through translation, and the common basis is used only to read out the coordinates of that function in the Hilbert space. We agree that this form is clearer than writing the shift on the basis side, and we will revise the manuscript accordingly.
> > >
> > > Eq. (4) is therefore best understood not as the most general Hilbert-space expansion, but as a translation-based representation choice motivated by the translation-invariance assumption. Under this representation, the cross-correlation form is derived only afterward in Eq. (5), via Parseval’s identity and a change of variables. In this sense, Eq. (4) specifies how positional information enters the representation, rather than directly presupposing the later conclusion.
> > >
> > > We thank the reviewer again for identifying this point, as it concerns the core starting point of the derivation and has helped us make the logical structure of our presentation clearer.

---

### Official Review · Reviewer_FWgu · 2026-03-12

**Soundness:** 3
**Presentation:** 3
**Significance:** 3
**Originality:** 2
**Overall Recommendation:** 5
**Confidence:** 3

**Summary:**

The paper proposes a theoretically justified way to set the n-dimensional frequencies in RoPE when applied over n-dimensional domains.
Precisely, instead of sampling the components of frequency vectors along each of the $n$ axes independently, they suggest a method to jointly sample the frequency vectors to ensure better coverage of the frequency spectrum and maximum symmetry of the samples set.
The authors empirically validate the benefits of this strategy with experiments on 2D and 3D datasets, ablation studies and out-of-domain generalization experiments.

**Compliance With Llm Reviewing Policy:**

Affirmed.

**Key Questions For Authors:**

My main confusion is due to the observation that applying a nD-RoPE rotation $e^{j \boldsymbol{\omega}^T \boldsymbol{x}}$ to a feature is equivalent to apply standard RoPE $n$ times, i.e. $e^{j \omega_1 x_1} \cdot \dots \cdot e^{j \omega_n x_n}$.
Do I understand correctly that this is the formulation used in the axial RoPE baselines?
Is then correct to say that the main contribution is sampling the frequency vectors $\boldsymbol{\omega}$ in a (smarter and theoretically justified) joint way, rather than sampling each axis-aligned component $\omega_i$ independently?

The argument in Sec. 3.2 seems interesting but I think could use some further elaboration. Why is it a problem if RoPE-Mixed learns frequency vectors not covering the spectrum completely? Why should this limit generalization and make the representation unstable? Is this somehow based on a symmetry assumption that all directions are equally important in the data?
This is also reflected in the beginning of Sec. 4.2: why is maximal symmetry, promoting invariance under rotations and reflections, important? I understand how that can be a good initialization choice maybe for RoPE-Mixed, but why would it be a bad if RoPE-Mixed deviated from this property after training?

The derivation in 4.1 are really nice to read. It could be useful to include some simple visualization of Eq. 13.

Sec. 4.2 "coverage": this paragraph is a bit confusing: it is not clear where the constraint in Eq. 14 comes from and why the system Eq. 15 is built? These constructions seem to be just taken from granted somehow.

Sec. 4.2 "maximum symmetry": the placement of the M=n+1 vectors is also a bit unclear. After some thought I understand this is just placing a (randomly rotated?) n-simplex in $\mathbb{R}^n$, but maybe it could still be made more visual in Sec 4.2.
Apart from this, I am still confused about the claim this arrangement achieves maximal isotropy and that M>n+1 leads to anisotropy. In 2D I think about M=3=2+1 points laid at 120 degrees angle forming the hexagonal lattice you mentioned yourself. But M >> n+1 points can be still placed uniformly in a circle achieving symmetry under rotations by $2\pi/M$; in 3D, one could use the vertices of regular polyhedrons. Wouldn't that be better?


Table 3-4: what attention is used for other baselines? Is it fair to compare nD-RoPE (vector attention) with other baselines or should we look instead at the (standard dot-product) row? In particular, in Tab. 4 nD-RoPE (standard dot-product) performs worst than all baselines. Apx D.4 mostly addresses this question, but I think Sec 5.1 could include at least a comment about this.


Minor comment: Sec. 1, lines 010-012: "transformer architecture itself does not preserve the order of input positions unless..." That seems badly phrased: attention preserves the order since it is permutation invariant; the problem is that it ignores positional information.

**Limitations:**

The paper doesn't explicitly discuss limitations of the proposed method, but includes a few useful ablation studies.

**Strengths And Weaknesses:**

I am not fully familiar with the related works on RoPE for n-dimensions (hence, I might be missing some context), but the manuscript is well written, has interesting theoretical justifications and comes with extensive empirical validation, showing the benefits of the proposed method.
I found Sec 4.1 particularly interesting, giving a useful justification for the use of RoPE in transformers (I wonder if this is already a well known result, but in any case it's an interesting addition to the paper).

Finally, the presentation is generally very clear, although a few parts could use some further explanation (see questions below).

---

> ### Author Rebuttal · Authors · 2026-03-30
>
> Thank you for the careful reading and thoughtful questions. Our responses are as follows. Additional figures are available at this anonymous [supplementary link](https://anonymous.4open.science/r/nD-RoPE-26B9/reb.pdf).
>
> **[Q1] Joint Wave-Vector Design vs. Axis-Aligned RoPE**
>
> Broadly yes, but with an important distinction. In axial RoPE, each
> frequency vector $\omega$ is associated with a single coordinate axis, and the final encoding is formed by stacking these axis-wise components, so the interaction between $\omega$ and $x$ remains separable across dimensions. By contrast, our method treats both $\omega$ and $x$ as full vectors, so each wave vector encodes information jointly across coordinates. Thus, one key contribution is the principled joint design of $\omega$, which leads to a fundamentally different spatial representation from axial RoPE.
>
> **[Q2] How Wave-Vector Sampling Affects Coverage and Symmetry in Real
> Space**
>
> The distribution of frequency vectors $\omega$ in **reciprocal space** determines the coverage and symmetry in **real space**. If the learned $\omega$ vectors cluster in a few directions or low-frequency regions, the representation becomes directionally imbalanced.
>
> Incomplete or anisotropic spectral coverage biases the encoding, weakens undercovered directions, and can reduce robustness to transformations such as rotation. Highly redundant or nearly collinear frequency vectors also make the representation ill-conditioned and sensitive to perturbations.
>
> Our argument does not assume all directions are equally important. Rather, maximal symmetry serves as a well-conditioned prior against anisotropic collapse under unconstrained parameterization. We will clarify this point in Secs. 3.2 and 4.2.
>
> **[Q3] Coverage Criterion**
>
> We agree the coverage criterion is presented too abruptly. Our goal is to explain how wave vectors $\omega$ in reciprocal space induce coverage in real space, since positional representations should span the full space and avoid unconstrained directions. See the anonymous [figure](https://anonymous.4open.science/r/nD-RoPE-26B9/reb.pdf) for an illustration.
>
> Eq. (14) formalizes the periodic phase structure induced by a single Fourier mode. Geometrically, the condition $\omega_i^T x = 2\pi k_i$ defines a family of equal-phase hyperplanes repeated along the direction of $\omega_i$. Eq. (15) expresses these constraints in matrix form, allowing us to analyze whether they jointly span the full real space. In particular, the rank condition $rank(\Omega)=n$ characterizes when the representation is full-dimensional and non-degenerate: if the rank is deficient, some spatial directions remain unconstrained. It also indicates the minimal number of wave vectors required.
>
> **[Q4] Maximum Symmetry Criterion**
>
> We agree our wording in Sec. 4.2 was imprecise. Here, *maximal symmetry* refers to the symmetry of the induced lattice in real space, rather than the directional arrangement of wave vectors in reciprocal space.
>
> Thus, we do not claim the simplex is globally optimal over all choices of $M$, but to identify it as the most isotropic construction under a minimal full-dimensional design. This is motivated by three criteria: full coverage in real space,
> isotropy (no preferred directions), and minimality, which preserves
> representational budget for multiple scales. Under these requirements, the regular simplex is the natural choice: in n dimensions, it is the unique configuration of n+1 equal-norm vectors with **equal pairwise angles**, and provides a unified construction across dimensions.
>
> When $M>n+1$, highly symmetric directional sets may still exist in reciprocal space. However, this does not imply greater symmetry of the induced pattern in real space. In 2D, three 120° wave vectors already generate a hexagonal lattice, which is maximally symmetric among periodic 2D lattices. Adding more wave vectors may refine reciprocal-space sampling, but mainly introduces redundancy, since additional wave vectors can overlap or partially cancel rather than create new real-space symmetry.
>
> We will revise Sec. 4.2 to make these distinctions explicit.
>
> **[Q5] Attention Type Clarification**
>
> In Tables 3–4, our main comparison uses vector attention for all position embeddings, following the standard Point Transformer setting.
>
> We also include a standard dot-product attention variant to disentangle the role of positional embedding from that of the attention mechanism. In Tab. 3, nD-RoPE remains effective under this setting, showing that its benefit is not tied only to vector attention. In Tab. 4, the standard dot-product variant performs worse overall, which we interpret as indicating that on real-world point cloud data the attention architecture itself becomes a dominant factor. We will clarify this point in Sec. 5.1.
>
> **[Minor Comments]**
>
> We will revise Sec. 1 accordingly and add a simple visualization of Eq. 13.
>
> **[Limitations]**
>
> Please see our response to Reviewer `AdnE [Q3]`.

---

> > ### Author Rebuttal · Reviewer_FWgu · 2026-04-03
> >
> > I thank the authors for the detailed answers, which clarified most of my doubts and, hence, I maintain my very positive recommendation.
> >
> > Reading the reply to Reviewer fiBd, I noticed their point about the comparison with FoPE, of which I wasn't aware. While I see some valid points in Reviewer fiBd's argument, as far as I understand FoPE focuses on length extrapolation in RoPE in 1 dimension and, therefore, I don't think it really diminishes this work's originality.
> > Still, given my limited expertise on this topic, I maintain a low confidence in my review and I invite the authors to clarify this aspects with reviewer fiBd.

---

> > > ### Author Response · Authors · 2026-04-04
> > >
> > > We thank the reviewer for the positive assessment and sincerely appreciate the encouraging feedback!

---

### Decision · Program_Chairs · 2026-04-30

**Decision:**

Accept (regular)

**Comment:**

The paper received mostly positive reviews. Reviewers valued the principled and sound ideas, strong and convincing experiments. Concerns were raised about theoretical presentation, motivation and missing baselines and citations.

The rebuttals clarified most of the issues and I encourage the authors to incorporate the suggested changes in the next revision.

Reviewer fiBd was the only negative recommendation, mainly concerned with the omission of FoPE, which presents similar ideas. The authors added comparisons and clarified the relationship -- FoPE focus on extrapolation in the 1D case where the submission considers multiple dimensions. Reviewer fiBd was not convinced and still recommends rejection. Other reviewers discussed this particular issue and disagreed with fiBd. Thus, I follow the reviewer majority and recommend acceptance.